# InfoLaw: Information Scaling Laws for Large Language Models with Quality-Weighted Mixture Data and Repetition

**Weidong Zhou** [* 1] **Fengze Liu** [* 1] **Binbin Liu** [1] **Ping Guo** [1] **Zijun Wang** [2] **Bingni Zhang** [1] **Yifan Zhang** [1] **Yifeng Yu** [1] **Xiaohuan Zhou** [1] **Taifeng Wang** [1]

## Abstract

Upweighting high-quality data in LLM pretraining often improves performance, but in data-limited regimes, especially under overtraining, stronger upweighting increases repetition and can degrade performance. However, standard scaling laws do not reliably extrapolate across mixture recipes or under repetitions, making the selection for optimal data recipes at scaling underdetermined. To solve this, we introduce **InfoLaw** (Information Scaling Laws), a data-aware scaling framework that predicts loss from consumed tokens, model size, data mixture weights, and repetition. The key idea is to model pretraining as information accumulation, where quality controls information density and repetition induces scale-dependent diminishing returns. We first collect the model performance after training on datasets that vary in scale, quality distribution, and repetition level. Then we build up the modeling for information so that information accurately predicts those model performance. InfoLaw predicts performance on unseen data recipes and larger-scale runs (up to 7B, 425B tokens) with 0.262% mean and 0.959% max absolute error in loss, and it extrapolates reliably across overtraining levels, enabling efficient data-recipe selection under varying compute budgets.

## 1. Introduction

Training large language models (LLMs) requires access to high-quality data (Brown et al., 2020; Chowdhery et al., 2023). However, the availability of high-quality data is severely limited (Villalobos et al., 2024), and in the data-constrained settings, upweighting higher-quality data inevitably increases repetition, which has been shown to impair performance when excessive (Muennighoff et al., 2023). This issue is further exacerbated by the widespread adoption of overtraining (Touvron et al., 2023; Yang et al., 2025): a strategy that reduces inference costs compared to the compute-optimal regime (Hoffmann et al., 2022).

To address the shortage problem of high-quality data as model scale increases, a common compromise is to incorporate lower-quality data, thereby reducing the repetition of high-quality samples. Intuitively, high-quality data provides greater performance gains than low-quality data upon first exposure, but as repetition increases, the marginal benefit decays—eventually approaching that of unseen low-quality data. However, the optimal balance between quality and repetition remains unclear. A standard approach for identifying optimal mixing strategies is to run smaller-scale experiments and extrapolate their performance to larger compute budgets using scaling laws (OpenAI et al., 2024; Hoffmann et al., 2022; Chowdhery et al., 2023). Yet, as shown in Figure 1, under conditions of data repetition, standard scaling laws fail to reliably predict model performance at scale (Hernandez et al., 2022; Muennighoff et al., 2023). Moreover, they do not generalize across different mixing strategies, necessitating grid searches over data recipes—an approach that is costly even at small scales.

In this paper, we study the problem of scaling large language models in a data-aware regime, where training data consists of a heterogeneous mixture with varying quality levels, and each quality level is repeated to different extents. We introduce a theoretical framework, the InfoLaw, which accounts for both the scaling effects of mixture weights and the impact of repetition. Our formulation views training as a process of accumulating information from the training dataset, with model performance determined by the total information gained by the end of training. At each step, the information gain is modeled as the sum of contributions from different quality ranges. Within each quality range, the gain depends on two factors: an information density function, parameterized by quality (with higher quality assigned higher density), and an exponential decay term that

---

[*]Equal contribution [1]ByteDance [2]Computer Science and Engineering, University of California, Santa Cruz. Correspondence to: Fengze Liu <fengze.liu@bytedance.com>.

*Proceedings of the 43$^{rd}$ International Conference on Machine Learning*, Seoul, South Korea. PMLR 306, 2026. Copyright 2026 by the author(s).

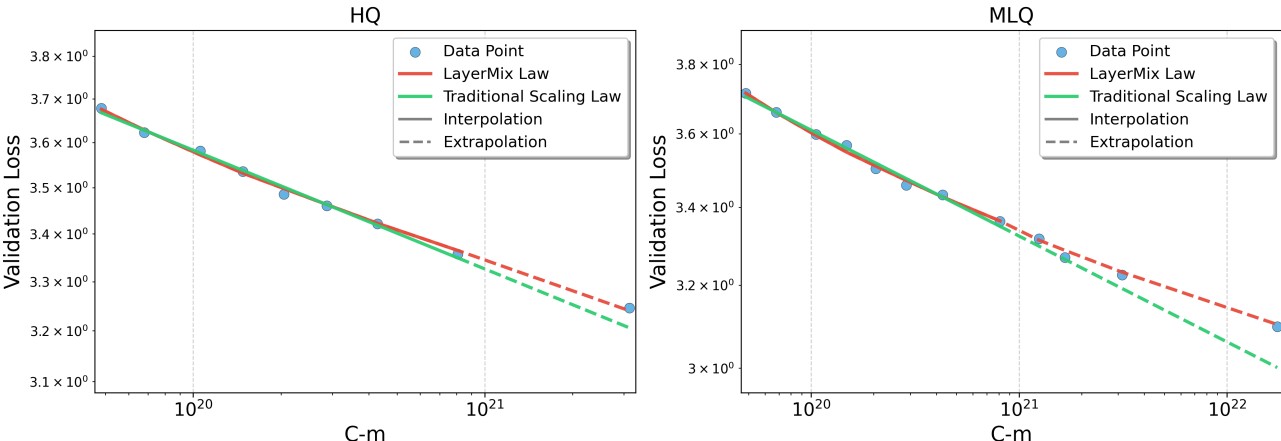

Figure 1. Validation loss versus compute $C_m$ in the loss-$C$ view under LayerMix data with repetition. Curves are fit on 252M-1.2B and extrapolated to larger models. The traditional scaling law mis-extrapolates under repetition, while InfoLaw tracks both interpolation and extrapolation across recipes (HQ and MLQ). We omit the MQ and MHQ recipes from this view for visual clarity; full loss-$C_m$ curves for all five presets are shown in Appendix Figures 8 and 9.

captures the interactions between model scale, data scale, and repetition level.

To fit the parameters of the InfoLaw, we construct a controlled suite of datasets that vary along three axes: scale, quality, and repetition level. We partition the source dataset into buckets according to document-level quality scores, and sample from these buckets with different weights, named as LayerMix sampling. Following the data-constrained setting, the source pool is downsampled to the target scale before sampling, so that repetition is induced in a stable and controlled way by the mixture weights. We then train 9 models ranging from 252M to 1.2B parameters from scratch, each under the same $3.6\times$ overtraining ratio (Gadre et al., 2024). For each model, we construct three datasets with distinct LayerMix sampling configurations, resulting in 27 total training runs that cover different quality-repetition trade-offs at each scale. Model performance is evaluated as the average validation loss across five downstream tasks. Finally, we fit the InfoLaw to these measurements, estimating the parameters that best capture the relationship between accumulated information and observed performance.

We evaluate the generalization of InfoLaw along three axes: (i) unseen mixture recipes (new LayerMix sampling weights), (ii) larger compute scales, and (iii) a higher overtraining ratio ($25\times$). Across these settings, InfoLaw accurately predicts loss on unseen recipes and scales up to a 7B model trained on 425B tokens, with 0.262% mean and 0.959% maximum absolute error. Moreover, using the fitted law we search over candidate mixtures and identify a data recipe for a 2.5B model that outperforms four randomly sampled baselines without additional training runs. The same parameters also extrapolate well to the $25\times$ overtraining regime.

## 2. Related Work

**Scaling Laws** Empirical studies have shown that transformer language models exhibit predictable power-law scaling with model size and training data (Hestness et al., 2017; Chowdhery et al., 2023; Radford et al., 2019), which has motivated the development of many large-scale systems, including dense models (Brown et al., 2020; Rae et al., 2021; Grattafiori et al., 2024) and mixture-of-experts variants (DeepSeek-AI et al., 2025; Yang et al., 2025; Fedus et al., 2021). Compute-based scaling laws further formalize how to allocate model capacity and training tokens under a fixed compute budget: Hoffmann et al. (2022) characterized the compute-optimal regime, while subsequent work explored alternative allocations and the interaction between compute $C$ and optimization choices such as batch size and learning rate (Kaplan et al., 2020; DeepSeek-AI et al., 2024).

In parallel, training smaller models on substantially more tokens than the compute-optimal point has become increasingly common for efficiency and deployment reasons (Touvron et al., 2023; Yang et al., 2025). Sardana et al. (2024) extended the Chinchilla framework by incorporating inference costs when deriving optimal training regimes, and Gadre et al. (2024) showed that scaling relations can remain reliable in overtrained regimes. For predicting downstream performance, Isik et al. (2025) studied how downstream metrics scale after fine-tuning, and Schaeffer et al. (2023) linked non-linear evaluation metrics to perplexity, supporting perplexity as a more stable proxy than earlier observations of emergent/unstable metrics (Wei et al., 2022).

**Data-Aware Scaling** Traditional scaling laws often assume effectively unlimited data, but in practice high-quality

data is scarce and therefore frequently upsampled (Lin et al., 2022). Under repetition, prior work reports diminishing returns and, beyond some point, performance degradation when upsampling subsets or repeating datasets (Hernandez et al., 2022; Muennighoff et al., 2023). At the same time, Xue et al. (2023) suggests that, in certain regimes, continuing to train on repeated data can still be preferable to stopping early, highlighting that the effect of repetition is non-trivial and not captured by classical laws. More recently, Chen et al. (2025) studied how scaling interacts with data density, providing a finer-grained view in limited-data regimes.

A separate line of work uses scaling laws to optimize data recipes. Ye et al. (2025) incorporated mixture weights into loss prediction, and Kang et al. (2025) argued that optimal mixing can be model-scale dependent. Liu et al. (2024) uses proxy models to search mixture ratios without training the full-scale model, while Gu et al. (2024); Que et al. (2024) leverage scaling insights in continued pre-training and domain-mixture design; Chang et al. (2024) further analyzes the interaction between scaling and data quality. Most closely related, Xie et al. (2023) reweights pretraining domains via group-DRO on a proxy model, Allal et al. (2025) curates a multi-stage domain mixture with manual refinement at each stage, and Shukor et al. (2026) derives a domain-aware scaling law that predicts loss as a function of model size, training tokens, and a domain weight vector. In contrast, our goal is to predict loss under quality-weighted mixtures with explicit repetition, enabling extrapolation across both mixture recipes and repetition levels.

## 3. Limitations of Conventional Scaling Laws

In this section, we reveal and substantiate a critical limitation of conventional scaling laws in the context of data repetition and quality selection. First, we introduce the LayerMix sampling function in section 3.1, to imitate real scenario where the data is a mixture of different quality and repetition degrees. Next, we compare the relationship between the model's loss $L$ and amount of compute $C$ in cases with and without repetition in section 3.2, and the results show that the traditional scaling law performs well on data without repetition

### 3.1. LayerMix Sampling Function

**Source Data**  We obtain our training corpora from Common Crawl (Common Crawl Foundation), following Penedo et al. (2023) and obtain 15T English tokens. We ran global fuzzy deduplication, a MinHash-based near-duplicate detection following Lee et al. (2022), across all 96 snapshots to ensure there is no repeat data in the corpora. This removed documents with high $n$-gram overlap, and the final dataset contains 3.7T tokens. Details are in Appendix B.

**Quality scoring and per-domain bucketing.**  We assign each document a quality score following Liu et al. (2025): we apply two quality classifiers: a fastText-based DCLM (Li et al., 2025) and a distilled-BERT FineWebEdu (Penedo et al., 2024), and take the average of their normalized scores. Applying both classifiers to the entire 3.7T-token corpus consumes approximately 64 H100 GPU $\times$ 16 hours. To prevent the bucket assignment from confounding quality with domain composition, we adopt a per-domain stratification, documents are first partitioned by their source domain, then ranked within each domain by the quality score, and finally split into six per-domain percentile buckets: 0-5%, 5-20%, 20-40%, 40-60%, 60-80%, 80-100%. Bucket $d$ then pools the same-percentile slices from all domains. Thus, bucket indices are quantitative (bucket 0 is the top-5% within each domain), while LayerMix weights $w$ preserve the domain mixture and isolate intra-domain quality redistribution.

**LayerMix sampling function.**  We define a LayerMix sampling function $H(w, K, S, B)$ to construct a packed training set; symbols are summarized in Appendix A. Here $S$ is the number of tokens in the source corpus to sample from, $K$ is the total number of tokens in the packed training set (we use one-epoch training to avoid additional epoch-induced repetition), $w = [w_0, \ldots, w_5]$ with $\sum_d w_d = 1$ specifies the target token proportions of the six buckets in the training set, and $B = [B_0, \ldots, B_5]$ specifies the bucket proportions in the source corpus (in our setting $B = [0.05, 0.15, 0.20, 0.20, 0.20, 0.20]$).

For bucket $d$, the training set contains $K_d = w_d K$ tokens sampled from $S_d = B_d S$ available source tokens. Let $M_d = \min(K_d, S_d)$ denote the number of unique (non-repeated) tokens from bucket $d$ that appear in the packed training set, and define the average repetition factor as

$$r_d = \frac{K_d}{M_d} = \frac{w_d K}{M_d},$$

so $r_d = 1$ when $K_d \leq S_d$ and $r_d > 1$ when the requested token count exceeds the available source pool. The full packing procedure is given in Algorithm 1 (Appendix D).

**Preset mixtures and search.**  By varying $(w, K, S)$, LayerMix produces datasets with different scale, quality mixture, and repetition. We enforce $w_d \geq w_{d+1}$ to keep higher-quality buckets more represented, and set $w_5 = 0$ to drop the lowest-quality 20% bucket. Unless stated otherwise, we set $K = S$ so that the resulting repetition pattern is induced purely by the weight vector $w$. To obtain the five preset weights in Table 1, we first randomly drew 30 candidate weight vectors satisfying the above constraints, then selected five that span distinct repetition regimes (HQ $\bar{r} \approx 4.35\times$, MHQ $3.33\times$, MQ $2.13\times$, MLQ $1.69\times$, LQ $1.35\times$). Section 5.2 subsequently shows that fitting Info-Law on the HQ/MQ/LQ subset already generalizes to the

*Table 1.* Preset LayerMix sampling weights and the optimal recipe searched by InfoLaw for a 2.5B model under $m = 3.6$. Each $w_d$ is the target token proportion of quality bucket $d$ in the packed training set; bucket 0 is the highest-quality 0-5% percentile (Section 3.1; see Appendix A for the full symbol list). The five preset recipes (HQ-LQ) span distinct repetition regimes (average per-bucket repetition factor $\bar{r}$ ranges from $4.35\times$ for HQ down to $1.35\times$ for LQ); the bottom row reports the recipe that minimizes InfoLaw-predicted loss over 100k random LayerMix candidates (Section 5.2). For larger model sizes and training-token budgets see Table 2.

| Name | $w0$ | $w1$ | $w2$ | $w3$ | $w4$ | $w5$ |
|------|------|------|------|------|------|------|
| **HQ (High Quality)** | 0.80 | 0.10 | 0.03 | 0.03 | 0.02 | 0.0 |
| **MHQ (Medium-High Quality)** | 0.66 | 0.22 | 0.05 | 0.03 | 0.02 | 0.0 |
| **MQ (Medium Quality)** | 0.48 | 0.23 | 0.13 | 0.07 | 0.07 | 0.0 |
| **MLQ (Medium-Low Quality)** | 0.38 | 0.21 | 0.20 | 0.11 | 0.08 | 0.0 |
| **LQ (Low Quality)** | 0.24 | 0.20 | 0.19 | 0.18 | 0.17 | 0.0 |
| **Optimal Recipe of 2.5B model with $m = 3.6$** | 0.50 | 0.49 | 0.01 | 0.0 | 0.0 | 0.0 |

held-out MHQ/MLQ presets and to 25 additional random weight vectors on a 1.2B model.

### 3.2. Traditional Scaling Law Between Loss and Amount of Compute

We compare the relationship between model loss $L$ and total compute $C$ under regimes with and without repetition in an overtrained setting. Specifically, under the compute-optimal scheme, $C_{opt} = N_{opt}K_{opt}$, where $K$ is the consumed tokens, $N$ is the non-embedding FLOPs per token as defined in DeepSeek-AI et al. (2024) and $N_{opt}$, $K_{opt}$ is the Chinchilla-optimal pair. Then in the overtrained setting, following Gadre et al. (2024), we set $K_m = \sqrt{m}K_{opt}$, $N_m = \frac{1}{\sqrt{m}}N_{opt}$, $C_m = K_mN_m$ with $m = 3.6$. And Gadre et al. (2024) shows that the the Loss–Compute relation preserves the fitted exponent for models trained with the same overtraining factor $m$.

The model loss is collected by training on datasets sampled with LayerMix parameters HQ and MLQ, see details in Table 1. Dataset HQ has more high quality data but with more repitition, while MLQ has more diverse data with less repitition. We then visualize the relationship between compute $C_m$ and model loss $L$ in the loss-$C_m$ view in Figure 1. Here $L$ is the average perplexity over five downstream tasks—HellaSwag (Zellers et al., 2019), ARC-E/ARC-C (Clark et al., 2018), MMLU (Hendrycks et al., 2021), and TriviaQA (Joshi et al., 2017). Following Schaeffer et al. (2023), we convert downstream accuracies into perplexity to obtain a smoother scaling signal. As shown in Figure 1, although a conventional power-law scaling curve can interpolate within the fitting regime (252M-1.2B), it systematically mis-extrapolates as $C_m$ increases under LayerMix data with repetition, yielding overly optimistic loss reductions. This failure appears consistently across representative mixture recipes, indicating that compute alone is insufficient to characterize scaling behavior in the presence of quality-weighted mixtures and repetition.

These observations suggest that traditional scaling laws are not reliably predictive under quality-weighted mixture data

with repetition, especially for extrapolation. Therefore, we need a modified scaling law that explicitly incorporates both the data quality distribution and the degree of data repetition as core variables.

## 4. Information Scaling Laws

In this section, we introduce the design of InfoLaw. We treat the training process as gaining information from the dataset and propose to calculate Information as accumulation of information gain throughout the training process, which synthesizes the impacts of data quality, repetition level, model scales and total training tokens, and then build power-law relationship with the model's final validation loss. Symbols introduced in this section are summarized in Appendix A.

### 4.1. Information Measurement

To build intuition for how repetition interacts with data quality, we compare two 850M runs trained with different Layer-Mix sampling weights. In the more repetition-heavy recipe (HQ), the top 5% quality bucket is repeated by roughly $16\times$, whereas in a less repetitive recipe (MQ) it is repeated by roughly $10\times$. Empirically, the two runs achieve similar evaluation loss early in training, but the more repetitive run improves substantially more slowly in the later stage and converges to a worse final loss, indicating diminishing returns from repeated exposures. See Appendix F Figure 5(b) for the training-time curves.

**Per-exposure gain on a single document.** Based on this observation, we propose an exponential decay function to model the diminishing information gain from repeated exposures of the same document. Let $I_i$ denote the intrinsic information content of a document $i$. We model the information that a language model extracts at the $t$-threpetition exposure of document $i$ as

$$I_{i,\text{part}}(t, \lambda(N)) = I_i \cdot \lambda(N)\, e^{-\lambda(N)t}, \quad (1)$$

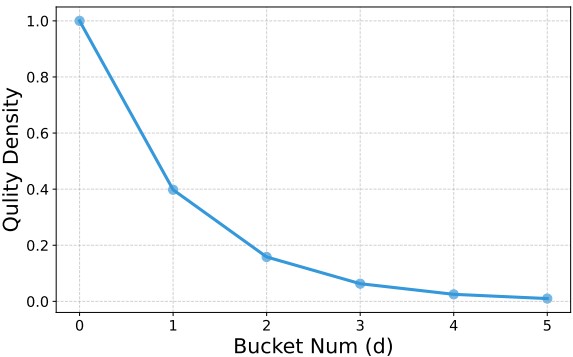

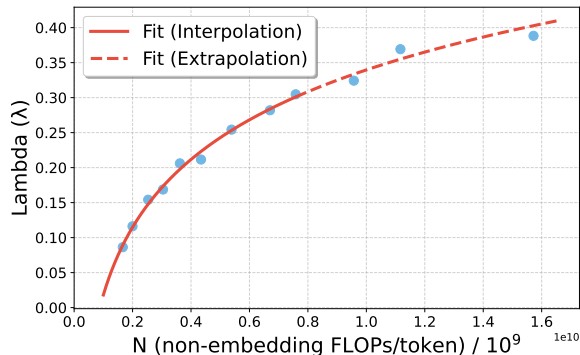

*(a)* The fitted quality density function $f_d$. The quality density is a monotonically decreasing function of the bucket index, meaning buckets with higher-quality data are assigned a higher density value.

*(b)* The relationship between $\lambda$ and $N$ with a fitted curve. The blue scattered points represent the observed data. The solid red line shows the fit within the data range, while the dashed line represents the extrapolation.

*Figure 2.* The fitted function of quality density function and relationship between $\lambda(N)$ and $N$

where $t$ is a positive integer indexing the repetition exposure (not a training step), and $\lambda(N) \geq 0$ is a rate parameter that depends only on the model's non-embedding FLOPs per token $N$. Intuitively, a larger model extracts information more efficiently per exposure and therefore saturates faster on repeated content, so we set $\lambda(N)$ to be an increasing function of $N$. We fit its functional form in Section 5.2; among constant, power-law, exponential, and logarithmic candidates, the logarithmic form $\lambda(N) = a \log N + b$ provides the best in-domain fit and extrapolation performance (Appendix H).

**All exposures of a single document.** When document $i$ is shown $T$ times in total, the total information the model extracts from it is obtained by integrating Eq. 1:

$$I_{i,\text{total}}(T, \lambda(N)) = \int_0^T I_{i,\text{part}}(t, \lambda(N))dt = I_i \cdot (1 - e^{-\lambda(N)T}) \quad (2)$$

Equation 2 captures the principle of diminishing returns: repeated exposure yields progressively smaller gains, and the total acquired information saturates asymptotically toward the document's intrinsic content $I_i$.

**Capturing the slowdown relative to total budget.** The plain exponential in Eq. 1 does not yet account for the empirical observation that the marginal benefit of repetition slows down further as the total training-token budget $K$ grows. We therefore incorporate a logarithmic normalization factor $\log K$:

$$I_{i,\text{part}}(t, \lambda(N), K) = I_i \cdot \lambda(N) e^{-\lambda(N)t/\log(K)} \quad (3)$$

The choice of $\log K$ rather than a constant or a power-law normalization is empirically grounded; we compare these

alternatives and justify the logarithmic form in Appendix C. Integrating the normalized rate over $T$ exposures gives

$$\begin{aligned} I_{i,\text{total}}(T, \lambda(N), K) &= \int_0^T I_{i,\text{part}}(t, \lambda(N), K) \, dt \\ &= I_i \log K \left(1 - e^{-\lambda(N)T/\log K}\right). \end{aligned} \quad (4)$$

**From a single document to the full training mixture.** Recall from Section 3.1 that quality bucket $d$ in the LayerMix-packed training set contains $K_d = w_d K$ tokens drawn from $M_d = \min(K_d, B_d S)$ unique source tokens, so each unique document in bucket $d$ is shown on average $r_d = K_d/M_d = w_d K/M_d$ times. The mapping $(w, K, S, B) \to \{r_d\}_d$ is therefore the sole channel through which the mixture weights and the resulting repetition pattern enter our Information functional. Substituting $T = r_d$ into Eq. 4 and summing the contributions across the six quality buckets gives the total Information learned by the model:

$$\begin{aligned} &\text{info}(w, K, S, f, \lambda(N)) \\ &= \sum_d I_d \left(1 - e^{-\lambda(N)r_d/\log(K)}\right) \\ &= \sum_d f_d M_d \log(K) \cdot \left(1 - e^{-\lambda(N)r_d/\log(K)}\right) \end{aligned} \quad (5)$$

where $d \in \{0, \ldots, 5\}$ indexes the six quality buckets, $I_d = f_d M_d \log K$ is the total intrinsic information packed in bucket $d$, and $f_d$ is a parameterized quality density function whose form is given in Section 5.2. Equation 5 factorizes into two interpretable parts: $I_d = f_d M_d \log K$ represents the information available in bucket $d$, and $\left(1 - e^{-\lambda(N)r_d/\log K}\right)$ represents the model's learning efficiency on this bucket when each document is on average repeated $r_d$ times.

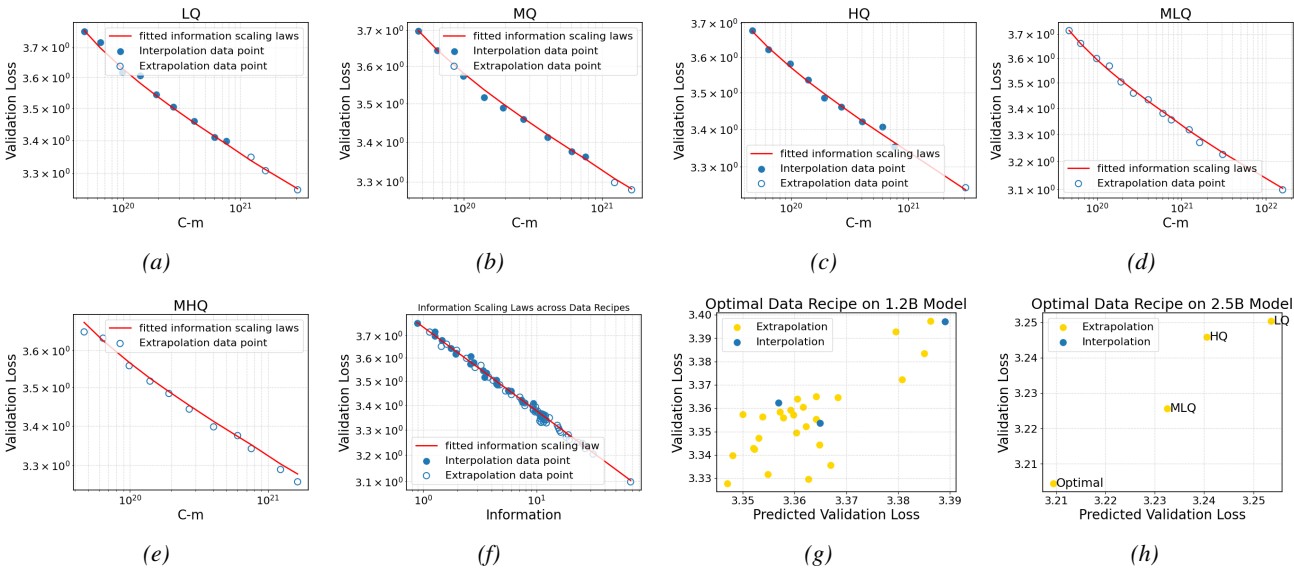

*Figure 3.* **Verification, Unification, and Application of Information Scaling Laws.** Panels **(a)-(e)** demonstrate that information scaling laws hold independently across varying data quality distributions (LQ to MHQ), consistently following power-law trajectories. **(f)** Illustrates the **Information Scaling Laws**, where diverse data recipes collapse onto a single curve when mapped to the information quantity metric, confirming the universality of the law. **(g)** Validates predictive capability on a 1.2B model, showing strong correlation between predicted and actual validation loss for both interpolation and extrapolation settings. **(h)** Demonstrates optimization on a 2.5B model, where the "Searched Optimal" recipe identified by our framework achieves lower validation loss compared to fixed baselines.

**Summary.** Information, parameterized by the LayerMix sampling weights $w$, the training-token budget $K$, the model scale $N$, and two fitted functions $(f, \lambda)$, quantifies the knowledge accumulated during training. Because it is designed to be monotonic with model performance, it lets us predict loss for arbitrary training configurations prior to any actual run. The fitting procedure for $f_d$ and $\lambda(N)$ is described in Section 5.2.

### 4.2. Information-Loss Power Law

As illustrated in Figure 1, conventional scaling laws in the loss-$C_m$ view are not reliably predictive under quality-weighted mixture data with repetition, with extrapolation errors that grow at larger compute. This motivates replacing compute with a repetition- and quality-aware effective data signal. Plotting the Information metric from Section 4.1 against validation loss $L$, Figure 3f shows that experimental points with different LayerMix sampling weights $w$, model FLOPs/token $N$, and training tokens $K$, collapse onto a single power-law curve. We accordingly model the relationship between $L$ and info as

$$L = \alpha \cdot \text{info}^{-\beta}, \tag{6}$$

with two scalar coefficients $(\alpha, \beta)$ fitted from data. In our experiments, $\alpha = 3.7373$ and $\beta = 0.0441$, so the law appears as a straight line of slope $-\beta$ and intercept $\log \alpha$ on a log-log plot.

**Free parameter count.** Despite the multi-stage derivation in Section 4.1, the InfoLaw formulation has only five free scalar parameters in total: $\theta$ in the quality density function $f_d = e^{-\theta d}$, $(a, b)$ in the rate function $\lambda(N) = a \log N + b$, and $(\alpha, \beta)$ in Eq. 6.

All five are jointly determined from the 252M-1.2B fitting experiments (Section 5.2), and no per-recipe or per-scale refitting is performed when extrapolating to unseen recipes, scales, or overtraining factors. As in classical scaling laws (Hoffmann et al., 2022), this lets us run inexpensive small-model experiments to compare data-recipe candidates and then use the fitted InfoLaw to extrapolate the performance of larger models trained on larger token budgets.

## 5. FITTING EXPERIMENTS

### 5.1. Training setup

We train 9 models ranging from 252M to 1.2B on 3 layermix sampling weights $HQ$, $MQ$, and $LQ$, with 3.6x over-trained ratio, resulting in 27 experiment runs in total to collect data for fitting the InfoLaw parameters. We use transformer architecture (Vaswani et al., 2017), SwiGLU (Shazeer, 2020) as the activation function and RoPE embeddings (Su et al., 2024). We use a tokenizer with 250k vocabulary. See Appendix D and Appendix E for details about LayerMix sampling weights, model structure, learning rate and optimizer.

*Table 2.* The best data recipe for different models and train token

| Model | Train Token | Source Token | $w0$ | $w1$ | $w2$ | $w3$ | $w4$ | $w5$ |
|-------|-------------|--------------|------|------|------|------|------|------|
| 7B | 300B | 500B | 0.548 | 0.444 | 0.004 | 0.003 | 0.002 | 0.000 |
| | 500B | 500B | 0.496 | 0.492 | 0.007 | 0.003 | 0.002 | 0.000 |
| | 800B | 500B | 0.439 | 0.430 | 0.130 | 0.001 | 0.000 | 0.000 |
| | 1000B | 500B | 0.395 | 0.387 | 0.214 | 0.003 | 0.001 | 0.000 |
| 1.8B | 300B | 500B | 0.619 | 0.376 | 0.004 | 0.001 | 0.000 | 0.000 |
| | 500B | 500B | 0.548 | 0.444 | 0.004 | 0.003 | 0.002 | 0.000 |
| | 800B | 500B | 0.496 | 0.492 | 0.007 | 0.003 | 0.002 | 0.000 |
| | 1000B | 500B | 0.491 | 0.487 | 0.017 | 0.005 | 0.000 | 0.000 |
| 1.2B | 300B | 500B | 0.758 | 0.229 | 0.012 | 0.001 | 0.000 | 0.000 |
| | 500B | 500B | 0.619 | 0.376 | 0.004 | 0.001 | 0.000 | 0.000 |
| | 800B | 500B | 0.496 | 0.492 | 0.007 | 0.003 | 0.002 | 0.000 |
| | 1000B | 500B | 0.496 | 0.492 | 0.007 | 0.003 | 0.002 | 0.000 |

## 5.2. Fitting the curve

We now describe how to use the 27 fitting runs (9 model scales × 3 LayerMix presets HQ/MQ/LQ as introduced in Section 5.1) to determine the five free scalar parameters of InfoLaw ($\theta$, $(a, b)$, and $(\alpha, \beta)$ from Section 4.2). The fitting proceeds in three stages: (i) determine $\theta$ and per-model $\lambda(N)$ values that maximize the rank consistency between Information and loss; (ii) fit a parametric $\lambda(N)$-vs-$N$ curve so $\lambda$ can be evaluated at unseen scales; (iii) fit the info-loss power-law coefficients $(\alpha, \beta)$ in Eq. 6. No per-recipe or per-scale refitting is performed thereafter.

**Stage 1: rank-based fit of $\theta$ and per-model $\lambda(N)$.**

$$(f^*, \lambda^*) = \underset{f, \lambda}{\operatorname{argmin}} \sum_{N, w} \rho_s\big(L_N, \operatorname{info}(w, K_N, S_N, f, \lambda(N))\big)$$
(7)

To make the search well-posed, we constrain $f$ to a one-parameter family enforcing monotonic decrease with bucket index $d$ (higher-quality buckets have larger density):

$$f_d(\theta) = e^{-\theta d}, \quad \theta > 0,$$
(8)

and we treat $\lambda$ as one scalar per model scale at this stage. Sampling 100,000 combinations of $\theta$ and the per-model $\lambda(N)$ values from the joint parameter space and minimizing Eq. 7 gives $\theta^* = 0.922$, with the fitted quality density $f(\theta^*)$ shown in Figure 2a.

**Stage 2: parametric fit of the $\lambda(N)$-$N$ curve.** With the per-model $\lambda(N)^*$ values from Stage 1, we now fit a parametric form so that $\lambda$ can be evaluated at unseen scales. Empirically the $\lambda(N)$-$N$ relation is non-linear: fast growth for small $N$ followed by gradual saturation (Figure 2b) and a logarithmic function can approximates well:

$$\lambda(N; a, b) = a \log(N/N_0) + b, \quad N_0 = 10^9.$$
(9)

Here $N$ is normalized by $N_0 = 10^9$ non-embedding FLOPs/token before applying the logarithm, making the logarithm dimensionless. Fitting Eq. 9 to the per-model $\lambda(N)^*$ values yields $a^* = 0.140$ and $b^* = 0.018$. We validate this fit by computing $\lambda(N)^*$ for larger $N$ under the fixed $\theta^*$ and checking whether these out-of-fit values lie on the predicted curve; Figure 2b shows that they do, supporting the extrapolation. Alternative forms and their relative fit and extrapolation quality are reported in Appendix H, and the logarithmic form is the best of the four.

**Stage 3: fit the info-loss power-law coefficients $(\alpha, \beta)$.** With $(f, \lambda)$ fixed at $\big(f(\theta^*), \lambda(N; a^*, b^*)\big)$, the Information metric is fully determined for any tuple $(w, K, S, N)$. We then fit the two scalar coefficients of Eq. 6 by linear regression in log-log space on the same 27 runs, obtaining $\alpha = 3.7373$ and $\beta = 0.0441$ (Section 4.2).

Together, $(\theta^*, a^*, b^*, \alpha, \beta)$ are the five free parameters that close the InfoLaw formulation. With them, we can predict the validation loss for any combination of LayerMix sampling weights $w$, training-token budget $K$, source-token budget $S$, and model FLOPs-per-token $N$ without any additional training run.

## 6. Extrapolation

After fitting InfoLaw on the 252M-1.2B models, we evaluate its extrapolation along three axes: unseen mixture recipes, larger model scales, and a higher overtraining ratio. Finally, we use our InfoLaw to predict optimal data recipe under different training budgets and validate the optimal recipe by comparing with preset recipes.

**Comparing with traditional scaling laws**

Figure 1 contrasts our InfoLaw with traditional power scaling law in the loss–$C$ plane. Both curves are fit using models in the 252M–1.2B range and then extrapolated to larger models. The Info curve tracks the MLQ data more closely within the fitting regime and remains accurate when extrapolating

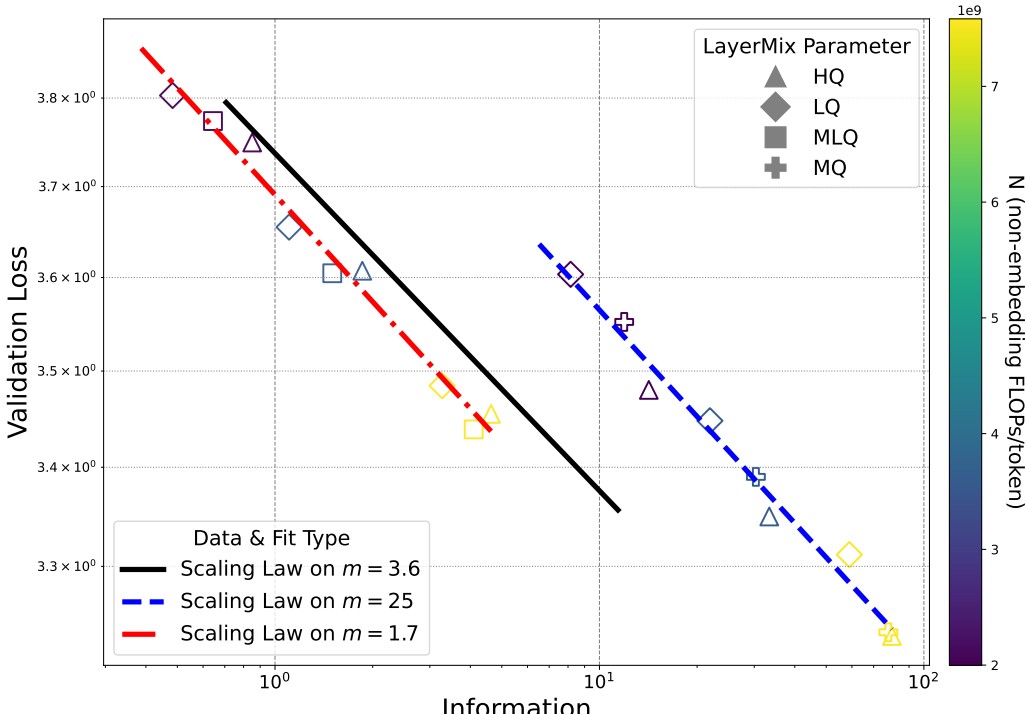

*Figure 4.* Cross-regime prediction of InfoLaw across three overtraining factors. Curves for $m = 3.6$, $m = 25$ and $m = 1.7$ are nearly parallel, supporting that the overtraining factor primarily shifts the intercept of the $L$-info power law while preserving its slope. All curves are evaluated with the same $f(\theta^*)$ and $\lambda(N)(a^*, b^*)$ fitted on $m = 3.6$. Marker shapes denote LayerMix recipes; color encodes $N$ (non-embedding FLOPs/token).

up to 7B models, avoiding the overly optimistic loss reductions predicted by the traditional law at high compute. Concretely, the traditional scaling law tends to under-estimate loss as $C_m$ grows, whereas the Info curve better matches the realized validation losses of larger models.

**Extrapolation to other LayerMix Sampling Weights**

We first test the ability to generalize to an unseen LayerMix sampling weights. We test on unseen dataset generated with $MLQ$, $MHQ$ on model scales ranging from $252M$ to $1.2B$, which are within the range of training data. Also we random sample 25 more sampling weights and run experiments on $1.2B$ model only.

The result is shown in Figure 3 . As can be seen, these points align remarkably well with the scaling law curve established by the initial $HQ$, $MQ$, $LQ$ data, demonstrating the predictive power of our model on unseen LayerMix sampling weights. The traditional scaling laws requires additional experiments on different data recipes to fit new curves, while ours can directly predict loss on unseen recipes.

**Extrapolation to Larger Models**

To test the extrapolation ability on model scale, we use the same Layermix sampling weights $MQ, LQ$ to train models ranging from 1.5B to 2.5B and $HQ$, $LQ$ to train model

with 2.5B parameters, which are out of the range of training data. The experimental results of larger models are shown in Figure 3(a-e), we can see InfoLaw predicts the loss on larger scale accurately for all three sampling weights, proving the ability of scaling on model size.

**Combination of Extrapolation**

Furthermore, we combine the two extrapolation above and test the effectiveness on both unseen LayerMix sampling weights and unseen scales. We run experiments with $MLQ$,$MHQ$ on models ranging from $1.5B$ to $7B$. As shown in Figure 3f, InfoLaw also generalise well on these combined extrapolation condition. On all the unseen data points, including unseen LayerMix sampling weights ($MLQ$, $MHQ$ and other 25 sets random sampled weights) and unseen model scales, InfoLaw predicts the validation loss with $0.262\%$ mean absolute relative error and maximum error is $0.959\%$. This proves that our proposed information scaling law has reliable extrapolation capability.

**Prediction error vs. experimental noise floor.** For each evaluation point we report the average of the last 3 checkpoints to reduce checkpoint-to-checkpoint noise; error bars are omitted from figures purely for visual clarity. To verify that InfoLaw's prediction error is not merely tracking ex-

perimental noise, we additionally estimate the experimental standard (SEM) at the out-of-distribution evaluation points (unseen recipes, unseen scales, and their combinations) following the noise framework of Heineman et al. (2026), obtaining mean SEM 0.189% and max SEM 0.887%. Info-Law's mean prediction error (0.262%) is therefore only $1.386\times$ the experimental SEM, indicating that the prediction is close to the measurement-precision floor of these benchmarks.

**Extrapolation to Different Overtrain Degrees**

To explore the reliability of InfoLaw under varying suboptimality, we conducted two additional experiment series at overtraining factors above and below the fitted $m = 3.6$ regime. The high-overtraining regime $m = 25$ was anchored by a 1.2B model trained on 640B tokens, contrasting with our initial $C_m$ experiment anchored at 106B tokens. The low-overtraining regime $m = 1.7$ is much closer to the compute-optimal point and complements the high-$m$ extrapolation by probing the opposite end of the overtraining spectrum.

For both new series, we computed the Information using the same quality density $f(\theta^*)$ and rate function $\lambda(N)(a^*, b^*)$ fitted on the $m = 3.6$ data. As shown in Figure 4, the three regimes each align with their own Info-loss power-law curve, and the three curves are nearly parallel. The fitted exponents $-\beta$ for $m \in \{1.7, 3.6, 25\}$ are $-0.0465$, $-0.0441$, and $-0.0463$, respectively, with mean absolute relative errors of 0.471%, 0.225%, and 0.408%, the full per-regime fit parameters are in Table 9. The exponent remains stable across these three regimes, which span more than an order of magnitude in $m$, suggesting that the overtraining factor primarily shifts the curve's intercept rather than its slope. This indicates that InfoLaw extrapolates reliably to overtraining factors well beyond the regime where it was fitted, both upward and downward.

**Optimizing Data Recipe with InfoLaw**

The ability of predicting loss on unseen data recipes and scales enables us to search for best data recipe without additional experiments. Similar to Liu et al. (2024). We randomly sample 100k LayerMix parameters from the parameter space, compute the information for each set of parameters, and convert it to loss via Equation 6. We then select the parameter that minimizes the predicted validation loss as the optimal LayerMix configuration for each training setting. To verify the optimal recipe, we conduct experiments on $2.5B$ model with optimal data recipe and 3 other Layermix sampling weights. The result optimal recipe is as in Table 1. As shown in Figure 3h, our optimal recipe achieves the best validation loss.

We additionally test generalization to unseen LayerMix parameters: on 25 held-out configurations for the 1.2B model,

predicted and measured validation losses achieve a Pearson correlation of 0.76, suggesting InfoLaw can reliably rank recipes for efficient search.

In Table 2, we present the optimal LayerMix parameters for different model sizes and training-token counts under a fixed source-token budget of 500B tokens. The optimal LayerMix parameters exhibit two clear trends. First, at a fixed training-token count, smaller models favor a higher fraction of high-quality data, whereas larger models benefit more from diversity and thus allocate a smaller fraction to the high-quality data. Second, as the total training tokens increase, the optimal LayerMix parameters shift from a high-quality emphasis toward greater diversity. More results are shown in Appendix K. In short: Small models or small training budgets prioritize quality; large models or large training budgets prioritize diversity.

# 7. Conclusion

In this paper, we propose a refined scaling law modeling **InfoLaw**, which focus on predicting model performance on downstream tasks under data-constrained settings with weighted-quality mixing. The InfoLaw provides accurate predictions of model performance on unseen data recipes at larger computational scales, achieving an mean absolute relative error of only 0.262% and a maximum error of 0.959%. This enables efficient discovery of optimal data recipes without the need for extensive additional experiments. Furthermore, the InfoLaw extrapolates reliably across varying degrees of over-training, offering an effective tool for selecting data recipes under different computational budgets.

# 8. Limitations

Our work has several limitations that suggest natural follow-up directions. First, the quality bucket boundaries are chosen heuristically; a systematic ablation over the number of buckets and their boundary placement could further improve predictive accuracy. Second, although we empirically observe that the overtraining factor $m$ primarily shifts the intercept of the info-loss curve while preserving its slope (Figure 4), we do not yet have a theoretical explanation for why this behavior holds across $m = 1.7$–25. Third, although per-domain bucketing makes the quality index quantitative within a fixed domain mixture, jointly modeling domain mixture, intra-domain quality, and repetition in a single scaling law remains open. We view this as a promising direction for unifying our work with domain-level data-aware scaling laws such as Shukor et al. (2026).

## Impact Statement

This paper aims to advance machine learning by improving our understanding of large language model performance under different data mixing and repetition strategies. Our InfoLaw can support more efficient pretraining by reducing expensive trial-and-error over data recipes. We do not anticipate direct negative societal consequences arising uniquely from this contribution. Broader ethical issues associated with LLMs, such as bias, misuse, and unsafe deployment, remain important but are not specifically introduced or materially amplified by our method beyond general improvements in training efficiency.

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

## A. Notation

Table 3 summarizes the symbols used throughout the paper. Symbols are grouped by the section in which they are first introduced.

*Table 3.* Notation used throughout the paper. Numerical values in parentheses are the fitted estimates obtained in Section 5.2.

| Symbol | Meaning |
|---|---|
| **Data quantities (Section 3.1)** | |
| $K$ | Total training tokens consumed by the language model. |
| $S$ | Total tokens in the source corpus before LayerMix sampling. |
| $w_d$ | Target token proportion of quality bucket $d$ in the packed training set, with $\sum_d w_d = 1$. |
| $B_d$ | Source proportion of quality bucket $d$; in our setting $B = [0.05, 0.15, 0.20, 0.20, 0.20, 0.20]$. |
| $K_d$ | Tokens drawn from bucket $d$, $K_d = w_d K$. |
| $M_d$ | Unique (non-repeated) tokens from bucket $d$, $M_d = \min(K_d, B_d S)$. |
| $r_d$ | Average repetition factor for bucket $d$, $r_d = K_d / M_d \geq 1$. |
| $t$ | Index of the $t$-th repetition exposure of a single document (Eq. 1); not a training step. |
| **Model scale and overtraining (Section 3.2)** | |
| $N$ | Non-embedding FLOPs per token of the model; used as a model-size proxy following DeepSeek-AI et al. (2024). |
| $m$ | Overtraining factor; $m = 1$ recovers the Chinchilla-optimal regime. |
| $K_m, N_m, C_m$ | Tokens, FLOPs-per-token, and compute under overtraining factor $m$: $K_m = \sqrt{m}\, K_{\text{opt}}$, $N_m = N_{\text{opt}}/\sqrt{m}$, $C_m = K_m N_m$. |
| **Information functional (Section 4.1)** | |
| $I_i$ | Intrinsic information content of document $i$ (Eq. 1). |
| $I_d$ | Total intrinsic information packed in bucket $d$, $I_d = f_d M_d \log K$ (Eq. 5). |
| $f_d$ | Quality density of bucket $d$; constrained to $f_d = e^{-\theta d}$ (Eq. 8). |
| $\theta$ | Quality density slope; controls how fast $f_d$ decays with bucket index $d$ (fitted: $\theta^* = 0.922$). |
| $\lambda(N)$ | Repetition decay rate; depends on model scale via $\lambda(N) = a \log(N/N_0) + b$, where $N_0 = 10^9$ non-embedding FLOPs per token (Eq. 9). |
| $a, b$ | Coefficients in $\lambda(N) = a \log(N/N_0) + b$, where $N_0 = 10^9$ non-embedding FLOPs per token. |
| info | Total Information learned by the model during training (Eq. 5). |
| **Loss and power law (Section 4.2)** | |
| $L$ | Validation loss, computed as average perplexity over five downstream tasks (HellaSwag, ARC-E/C, MMLU, TriviaQA). |
| $\alpha, \beta$ | Info-loss power-law coefficients in $L = \alpha \cdot \text{info}^{-\beta}$ (fitted: $\alpha = 3.7373$, $\beta = 0.0441$). |

## B. Training Dataset

We use the English portion of the Common Crawl Dataset (Common Crawl Foundation), utilizing 96 of the snapshots, from CC-MAIN-2013-20 to CC-MAIN-2024-18. We then ran MinHash-based global fuzzy deduplication (Lee et al., 2022) across all snapshots to remove documents with high $n$-gram overlap, yielding a final dataset with 3.7T tokens.

## C. Justification for the Normalization Term $\log(K)$

In Equation 3, we incorporate a normalization term $\log(K)$ into the decay function to model the interaction between repetition decay and the total token budget. We selected this logarithmic form after rigorously evaluating alternative formulations. Specifically, we compared our chosen decay term against constant normalization and power-law normalization:

- **Constant Normalization:** Assuming the decay rate is independent of the dataset scale:

$$\text{Decay}(t) \propto e^{-\lambda(N)t} \tag{10}$$

- **Power-Law Normalization:** Assuming the decay scales polynomially with the token budget:

$$\text{Decay}(t) \propto e^{-\frac{\lambda(N)t}{K^\alpha}} \tag{11}$$

- **Logarithmic Normalization (Ours):**

$$\text{Decay}(t) \propto e^{-\frac{\lambda(N)t}{\log(K)}} \tag{12}$$

While we omit the visual plots for brevity, our preliminary experiments demonstrated that the alternative forms failed to unify the scaling behaviors across different token budgets $K$:

1. **Failure of Constant Normalization:** This formulation fails to account for the scaling properties of information density. Empirically, we observed that it systematically overestimates the accumulated Information for large models when trained with larger token budgets. Consequently, this leads to overly optimistic loss predictions that deviate significantly from the actual experimental results.

2. **Failure of Power-Law Normalization:** We found this formulation to be fundamentally unsuitable. It resulted in a complete failure to fit the relationship between Information and Validation Loss. The data points derived using power-law normalization remained scattered without exhibiting the necessary power-law correlation, rendering it impossible to derive a valid scaling law.

In contrast, the $\log(K)$ term was the only formulation that minimized the alignment error—successfully collapsing diverse configurations of $(w, K, S)$ onto a single unified power-law curve (as shown in Figure 3f)—and maintained a low extrapolation error across the full range of model scales (252M to 7B). This suggests that the marginal utility of repeated data diminishes logarithmically relative to the total training budget.

## D. LayerMix Sampling Function

We show the detail of LayerMix sampling function in Algorithm 1.

$$\sqrt{m} = \frac{N_{opt}}{N} = \frac{D}{D_{opt}} \tag{13}$$

A value of $m = 1$ indicates a compute-optimal training run, while $m > 1$ signifies that the model is overtrained relative to its compute budget.

## E. Training

The model structures used in LayerMix are illustrated in Table 4. We train all the model with 2048 as the max sequence length, we use a cosine decay scheduler and the initial learning rate calculated by $lr = round(0.3118 \cdot C^{-0.1250}, 8)$, the warm up ratio is set 0.5%. We use AdamW optimizer with $\beta_1 = 0.9$, $\beta_2 = 0.95$, weight decay$= 0.1$.

## F. Supplementary Analysis of Repetition Effects

**Notation.** IST (Infinite Source Tokens) denotes $S \gg K$, where repetition is negligible; LST (Limited Source Tokens) denotes $S = K$, where repetition is induced by the sampling weights. HQ/MQ refer to the LayerMix preset recipes in Table 1. Figure 5(a) provides an additional sanity check for the loss-$C_m$ behavior under different repetition regimes, while Figure 5(b) shows the corresponding training-time dynamics motivating a saturation/decay model.

## G. The relationship between benchmark validation loss and performance

Our InfoLaw focus on predicting the evaluation loss on downstream benchmarks. However, it also represents for the actual downstream performance. Figure 6 shows a near-linear relationship between validation loss and downstream performance on our evaluation tasks, and Table 5 shows the spearman corelation between validation loss and downstream performance. Lower loss consistently corresponds to higher performance within the operating regime of our models. This indicates that improvements in loss provide reliable signals for expected gains in downstream performance.

---

**Algorithm 1** LayerMix Sampling Function $H(w, K, S, B)$

---

1: **function** H$(w, K, S, B)$

**Require:**

$w$: A list of target proportions for six buckets, $w = [w_0, ..., w_5]$, where $\sum w_d = 1$.

$K$: The total number of tokens for the final training dataset.

$S$: The total number of tokens in the entire source corpora.

$B$: The source distribution proportions B=[0.05, 0.15, 0.2, 0.2, 0.2, 0.2].

**Ensure:**

$D_{train}$: The final packed training dataset.

$M$: A list of unique token counts for each layer, $M = [M_0, ..., M_5]$.

$R$: A list of average repetition counts for each layer, $r = [r_0, ..., r_5]$.

2:      Initialize an empty training dataset $D_{train} \leftarrow \emptyset$.

3:      Initialize empty lists for statistics: $M \leftarrow [], r \leftarrow []$.

4:      **for** $d \leftarrow 0$ **to** 5 **do**                              ▷ Iterate through each quality bucket

5:          $K_{needed} \leftarrow K \times w_d$         ▷ Calculate tokens needed from bucket $d$ for the target mix

6:          $S_d \leftarrow S \times B[d]$              ▷ Calculate source tokens available in bucket $d$

                                              ▷ Calculate the sampling ratio for the current bucket

7:          $Ratio_d \leftarrow K_{needed}/S_d$

                                              ▷ — Detailed sampling process for bucket $d$ —

8:          Initialize an empty temporary set $D_{sampled\_d} \leftarrow \emptyset$.

9:          **for all** data point $x$ in bucket $d$ **do**

                                           ▷ 1. Deterministic copy for the integer part of the ratio

10:              **for** $i \leftarrow 1$ to $\lfloor Ratio_d \rfloor$ **do**

11:                  Add $x$ to $D_{sampled\_d}$

12:              **end for**

                                           ▷ 2. Probabilistic sampling for the fractional part

13:              **if** $Ratio_d - \lfloor Ratio_d \rfloor > 0$ and $random() < (Ratio_d - \lfloor Ratio_d \rfloor)$ **then**

14:                  Add $x$ to $D_{sampled\_d}$

15:              **end if**

16:          **end for**

17:          Append all data from $D_{sampled\_d}$ to $D_{train}$.

18:          $M_d \leftarrow \min(K_{needed}, S_d)$          ▷ Calculate unique tokens based on the new formula

19:          Append $M_d$ to $M$.

20:          $r_d \leftarrow K_{needed}/M_d$               ▷ Calculate average repetition count

21:          Append $r_d$ to $R$.

22:      **end for**

23:      **return** $D_{train}, M, r$                          ▷ Return dataset and statistics

24: **end function**

---

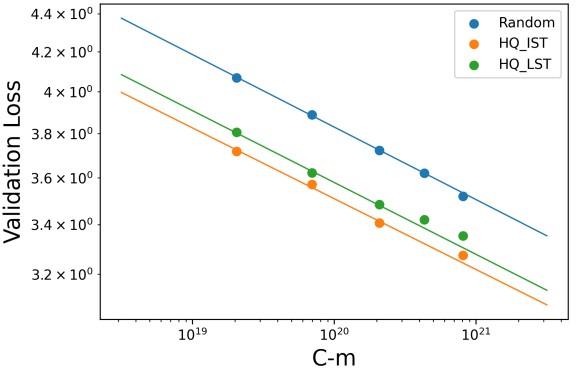

*(a)* Loss-$C_m$ curves under different data regimes. Random: large source ($S \gg K$) with negligible repetition. HQ_IST: LayerMix with the HQ recipe and $S \gg K$ (negligible repetition). HQ_LST: the same HQ recipe but $S = K$, inducing repetition.

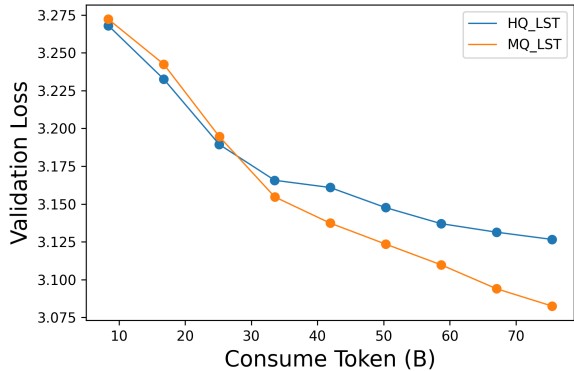

*(b)* Training-time evaluation loss for two 850M runs (HQ_LST vs MQ_LST), illustrating late-stage slowdown under heavier repetition.

*Figure 5.* **Supplementary evidence for repetition effects.** (a) In the loss-$C_m$ view, repetition induces systematic deviation from a single power-law trend. (b) Heavier repetition leads to slower late-stage improvement and worse final loss, consistent with diminishing returns.

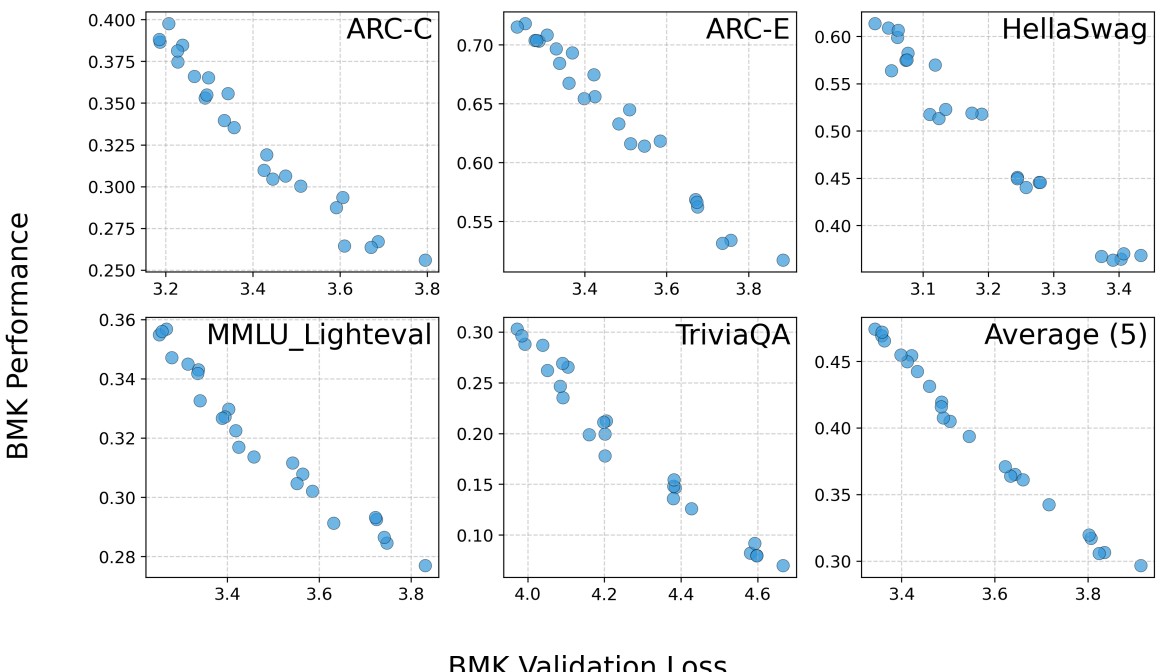

*Figure 6.* Validation loss versus downstream performance across benchmarks (ARC-C, ARC-E, HellaSwag, MMLU-Lighteval, TriviaQA) and their average.

---

**Algorithm 2** Calculation of Overtrain Degree and Optimal Tokens

---

1: **function** CALCULATEOVERTRAINEXTRAPOLATION($model_{curr}, D_{curr}, models_{target}$)

**Require:**

    $model_{curr}$: The size of the current model configuration.

    $D_{curr}$: The number of tokens used to train the current model.

    $model_{target}$: The size of the target model configuration.

**Ensure:**

    $m$: The calculated overtrain degree for the current configuration.

    $D_{target}$: The train token of target model under same overtrain degree.

2:    *// Part 1: Calculate overtrain degree m from the current configuration*

3:    $N_{curr} \leftarrow \text{Get\_N}(model_{curr})$     ▷ Get $N$ (non-embedding FLOPs/token) for the current model

4:    $C \leftarrow N_{curr} \times D_{curr}$     ▷ Calculate the total compute budget

5:    $N_{opt} \leftarrow 0.06085 \times C^{0.5445}$   ▷ Calculate Chinchilla-optimal model non-embedding FLOPs/token for budget $C$

6:    $D_{opt} \leftarrow 16.4326 \times C^{0.4555}$     ▷ Calculate Chinchilla-optimal tokens for budget $C$

7:    $\sqrt{m} \leftarrow N_{opt}/N_{curr}$     ▷ Calculate the overtrain degree $m$

8:    ▷ This is equivalent to $\sqrt{m} = D_{curr}/D_{opt}$

9:    *// Part 2: Extrapolate to target model while keeping m constant*

10:    **for** each $model_t$ in $[model_{curr}] + models_{target}$ **do**

11:        $N_t \leftarrow \text{Get\_N}(model_t)$     ▷ Get $N$ (non-embedding FLOPs/token) for the target model

12:        $N'_{opt} \leftarrow N_t \times \sqrt{m}$   ▷ Find the corresponding optimal model non-embedding FLOPs/token for the target

13:        $C_{new} \leftarrow (N'_{opt}/0.06085)^{1/0.5445}$     ▷ Derive the new compute budget

14:        $D'_{opt} \leftarrow 16.4326 \times C_{new}^{0.4555}$     ▷ Find optimal tokens for the new budget

15:        $D_{target} \leftarrow D'_{opt} \times \sqrt{m}$     ▷ Calculate the required tokens for the target model

16:    **end for**

17:    **return** $m, D_{target}$     ▷ Return the overtrain degree and the train token of target model under same $m$.

18: **end function**

---

## H. Alternative Fits for $\lambda$

In Section 5.2, we model the relationship between non-embedding FLOPs/token $N$ and hyperparameter $\lambda$. Our primary specification adopts the logarithmic form Equation 9. Beyond this baseline, we also evaluated alternative function families, including an exponential form:

$$\lambda(x; a, b, c) = a \cdot \left(1 - e^{-bx+c}\right) \tag{14}$$

and a power-law form:

$$\lambda(x; a, b) = a \cdot x^b \tag{15}$$

As shown in Figure 7, the logarithmic model achieves the best fit to the $N - \lambda$ relationship, outperforming the exponential and power-law alternatives. Accordingly, we adopt function 9 as the final parameterization.

## I. Deviation of Traditional Scaling Law

We show all $Loss$-$C$ curve of different LayerMix sampling weights with IST and LST in Figure 8 and Figure 9, they all exhibit a clear deviation from the traditional scaling law, which is fitted from the first three data points.

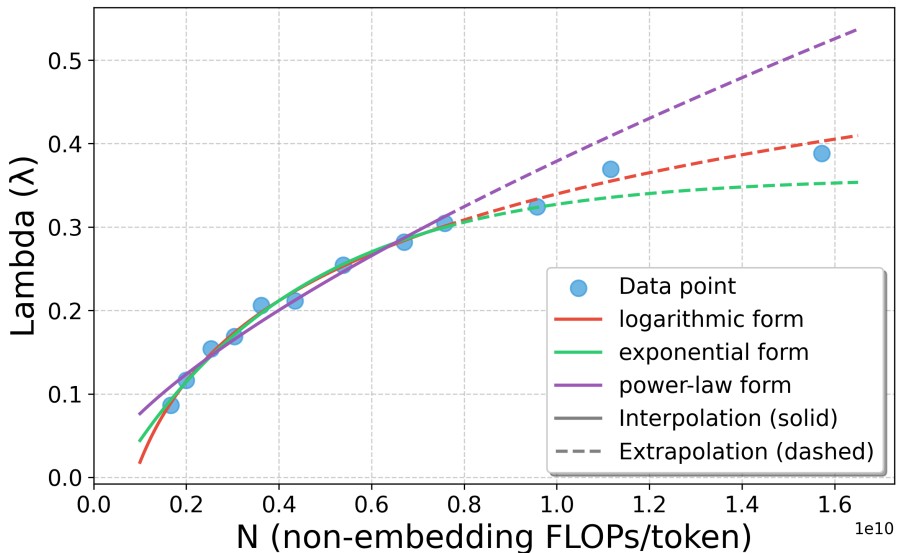

*Figure 7.* Comparison of functional fits for $\lambda$ as a function of $N$ (non-embedding FLOPs/token). The logarithmic form provides the best in-domain fit and extrapolation behavior compared with the exponential and power-law alternatives. Solid lines denote interpolation over observed $N$; dashed lines indicate extrapolation beyond the observed range.

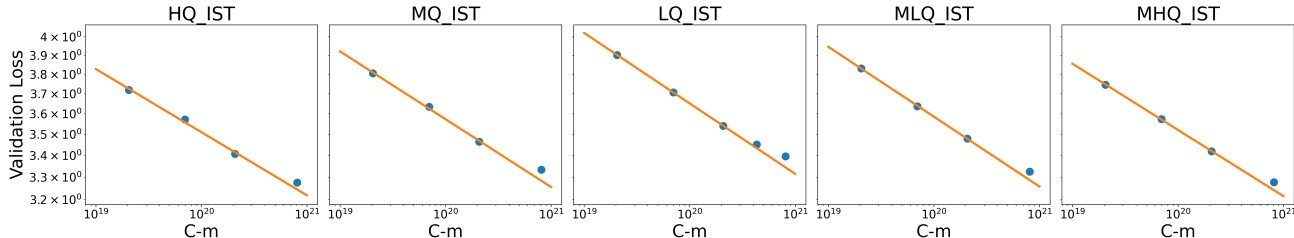

*Figure 8.* $Loss$ and $C_m$ Curve of different LayerMix $IST$ experiments

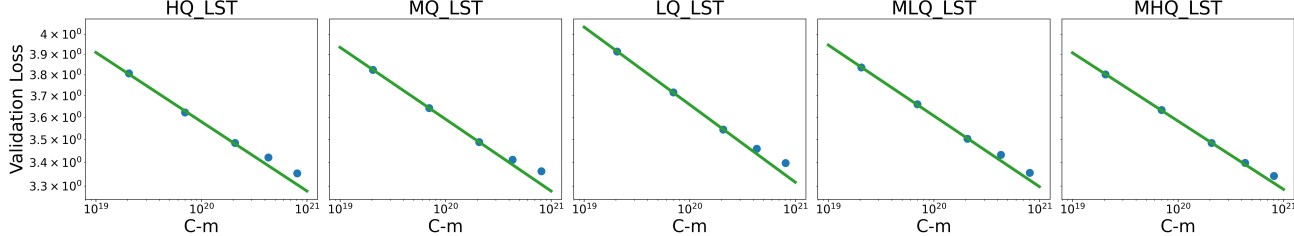

*Figure 9.* $Loss$ and $C_m$ Curve of different LayerMix $LST$ experiments

*Table 4.* Structure of models used in LayerMix.

| Model | Hidden dim. (C) | MLP dim. (D) | Layers (L) | Heads |
|-------|-----------------|--------------|------------|-------|
| **252M** | 1024 | 2752 | 20 | 16 |
| **302M** | 1024 | 2752 | 24 | 16 |
| **392M** | 1280 | 3392 | 20 | 20 |
| **470M** | 1280 | 3392 | 24 | 20 |
| **566M** | 1536 | 4096 | 20 | 24 |
| **680M** | 1536 | 4096 | 24 | 24 |
| **850M** | 1792 | 4800 | 22 | 28 |
| **1B** | 1920 | 5120 | 24 | 30 |
| **1.2B** | 2048 | 5440 | 24 | 16 |
| **1.5B** | 2304 | 6144 | 24 | 36 |
| **1.8B** | 2304 | 6144 | 28 | 36 |
| **2.5B** | 2560 | 6848 | 32 | 40 |
| **7.7B** | 4096 | 14336 | 32 | 32 |

*Table 5.* Spearman correlation between validation loss and performance across benchmarks

| Benchmark | Spearman $r_s$ | $p$-value |
|-----------|----------------|-----------|
| ARC-C | -0.979 | $1.02 \times 10^{-16}$ |
| ARC-E | -0.982 | $2.72 \times 10^{-17}$ |
| HellaSwag | -0.942 | $6.13 \times 10^{-12}$ |
| MMLU-LightEval | -0.989 | $1.26 \times 10^{-19}$ |
| TriviaQA | -0.970 | $4.53 \times 10^{-15}$ |
| Average (5) | -0.996 | $3.54 \times 10^{-24}$ |

## J. Quality Score

We show some data samples in different Quality buckets in Table 6. This table indicates that high-score samples under our merged FineWebEdu and DCLM scores are more coherent and instructional. By contrast, low-score cases predominantly consist of advertisements or low-information content, offering little substantive value.

Table 7 reports four benchmark results for training a 1.2B model from scratch on 30B tokens using three datasets: the top 5% and top 20% selected by the FineWebEdu classifier, and a random sample, all from Penedo et al. (2023). High-quality data selected by FineWebEdu outperforms the random baseline, and higher-quality subsets yield better results.

## K. Optimizing Token Mix with InfoLaw

We present the detailed optimal LayerMix parameters (or token-mix ratios) for different models and training budgets predicted by InfoLaw in Table 8. This table shows that small models or small training budgets prioritize quality, while large models or large training budgets prioritize diversity.

## L. Generalization to Refinedweb

To evaluate the robustness and generalization capability of the InfoLaw across different data distributions, we conducted an additional series of verification experiments on the RefinedWeb dataset (Penedo et al., 2023).

**Experimental Setup.** We followed the identical data preprocessing, LayerMix sampling, and training procedures described in Section 3.1 and Section 5.1, with the sole exception of replacing the source corpus with RefinedWeb. Due to time and computational constraints, we limited the scope of this study to three LayerMix sampling configurations: HQ (High Quality) and LQ (Low Quality) were used for parameter fitting (interpolation), while MLQ (Medium-Low Quality) was held out for extrapolation testing. For each configuration, we trained models at three specific scales: 302M, 566M, and 1.2B parameters.

**Fitting and Extrapolation.** We applied the fitting methodology outlined in Section 5.2. Our analysis yielded two key observations:

*Table 6.* Case study contrasting data quality. Left column (Quality_Range 0–5%, top of the per-domain quality ranking) contains coherent, informational, and instructional passages. Right column (Quality_Range 80–100%, bottom of the ranking) contains low-information, ad-like content with minimal reasoning or educational value. Sentences are lightly edited for typography (escaped \n characters removed; verbose sections elided with "...").

|  | **Quality_Range** 0-5% **(top)** | **Quality_Range** 80-100% **(bottom)** |
|---|---|---|
| #1 | *Celebrate your way.* Whether you are having a picnic with your family, a barbeque with friends in the backyard or follow the Australian of the Year awards, Australia Day on the 26th of January is an occasion to come together as a nation and celebrate what's great about Australia. On Australia Day we celebrate the past, present and future of the country. It is a commemoration of the day that the First Fleet landed in Sydney Cove in 1788, as well as a celebration of all the achievements of our country... | *Re: BUY snes copier.* This message was posted by The Dumper on January 05, 2002 at 03:06:18 coming from 209.204.139. This message is a reply to "Will BUY snes copier" posted by Spongebob at 02:25:29. Looking for an snes copier. Preferably the Super Wildcard DX 2. Will pay $$$$$... |
| #2 | *Housing report, 7 March 2012 (Shirley Allen).* Monthly home prices in the United States increased by a seasonally adjusted 0.7% in December, following a similarly revised 0.7% gain in November according to the Federal Housing Finance Agency's (FHFA) monthly House Price Index (HPI). December's home prices were still 0.8% lower than a year ago; since the April 2007 market peak, home prices have declined over 18% and are at roughly the same levels last seen in March 2004... | *Wall-hanging product listing.* These adorable little wall hangings feature bright pops of colour, tassley texture and pretty little berry knots. How can you resist? Made from 100% recycled cotton and mounted on a Tasmanian Oak dowel. Measures approximately 16 cm wide by 33 cm long (including hanger). Colours vary from screen to screen. Custom orders are available—price available upon request. Outside Australia? Please contact us for country specific freight charges... |

*Table 7.* FineWebEdu-selected subsets vs. random data for training a 1.2B model on 30B tokens

| Model | Data | ARC-C | HellaSwag | TriviaQA | MMLU-LightEval | avg |
|---|---|---|---|---|---|---|
| 1.2B | Random 30B | 28.50% | 51.56% | 15.55% | 30.23% | 31.46% |
| 1.2B | FWE-top20% 30B | 34.30% | 55.26% | 20.05% | 32.82% | 35.61% |
| 1.2B | FWE-top5% 30B | 37.20% | 55.14% | 19.25% | 34.50% | 36.52% |

- **Consistency of Quality Density ($f$):** The fitted values for the quality density function $f_d$ were numerically very close to those derived from our primary dataset. Specifically, the fitted parameter $\theta$ is 0.93 for RefinedWeb, which is remarkably close to the value of 0.92 obtained from our primary dataset. We attribute this similarity to the fact that RefinedWeb (Penedo et al., 2023) is also derived from Common Crawl (Common Crawl Foundation); despite employing different filtering strategies, the shared underlying data source results in a comparable information density distribution.

- **Optimization of $\lambda(N)$:** In the main experiments, we modeled the relationship between the parameter $\lambda(N)$ and model scale $N$ using a logarithmic curve. However, due to the limited number of data points in this verification set (only three distinct model scales), fitting a robust $\lambda(N) - N$ curve was not feasible. Consequently, we skipped the curve fitting step for $\lambda(N)$ and directly searched for the optimal $\lambda$ values corresponding to the specific model sizes (302M, 566M, and 1.2B).

**Results.** Using the parameters fitted on the HQ and LQ configurations, we predicted the validation loss for the unseen MLQ configuration, as illustrated in Figure 11. The InfoLaw demonstrated strong predictive accuracy on the RefinedWeb dataset, achieving a maximum absolute error of 0.36% and a mean absolute percentage Error 0.24% on the extrapolated MLQ experiments. These results further corroborate that the InfoLaw effectively captures the fundamental trade-offs between data quality, repetition, and compute scale, independent of the specific underlying data source.

## M. Per-Regime Fit across Overtraining Factors

Table 9 reports the per-regime Info-loss power-law fits underlying Figure 4. All three curves share the same quality density $f(\theta^*)$ and rate function $\lambda(N)(a^*, b^*)$ that were fitted only on the $m = 3.6$ data; only the power-law coefficients $(\alpha, \beta)$ in $L = \alpha \cdot \text{info}^{-\beta}$ are refit per regime. The fitted exponents $-\beta$ stay within $[-0.0465, -0.0441]$ across $m \in \{1.7, 3.6, 25\}$, while $\alpha$ shifts by roughly an order of magnitude, consistent with our observation that the overtraining factor primarily moves

*Table 8.* The detailed best layer token mix for different models and train token

| Model | Train Token | Source Token | $w0$ | $w1$ | $w2$ | $w3$ | $w4$ | $w5$ |
|-------|-------------|--------------|------|------|------|------|------|------|
| 7B | 200B | 500B | 0.619 | 0.376 | 0.004 | 0.001 | 0.000 | 0.000 |
| | 300B | 500B | 0.548 | 0.444 | 0.004 | 0.003 | 0.002 | 0.000 |
| | 400B | 500B | 0.496 | 0.492 | 0.007 | 0.003 | 0.002 | 0.000 |
| | 500B | 500B | 0.496 | 0.492 | 0.007 | 0.003 | 0.002 | 0.000 |
| | 600B | 500B | 0.491 | 0.487 | 0.017 | 0.005 | 0.000 | 0.000 |
| | 700B | 500B | 0.439 | 0.430 | 0.130 | 0.001 | 0.000 | 0.000 |
| | 800B | 500B | 0.439 | 0.430 | 0.130 | 0.001 | 0.000 | 0.000 |
| | 900B | 500B | 0.404 | 0.403 | 0.183 | 0.006 | 0.003 | 0.000 |
| | 1000B | 500B | 0.395 | 0.387 | 0.214 | 0.003 | 0.001 | 0.000 |
| 1.8B | 200B | 500B | 0.825 | 0.165 | 0.005 | 0.004 | 0.001 | 0.000 |
| | 300B | 500B | 0.619 | 0.376 | 0.004 | 0.001 | 0.000 | 0.000 |
| | 400B | 500B | 0.548 | 0.444 | 0.004 | 0.003 | 0.002 | 0.000 |
| | 500B | 500B | 0.548 | 0.444 | 0.004 | 0.003 | 0.002 | 0.000 |
| | 600B | 500B | 0.496 | 0.492 | 0.007 | 0.003 | 0.002 | 0.000 |
| | 700B | 500B | 0.496 | 0.492 | 0.007 | 0.003 | 0.002 | 0.000 |
| | 800B | 500B | 0.496 | 0.492 | 0.007 | 0.003 | 0.002 | 0.000 |
| | 900B | 500B | 0.491 | 0.487 | 0.017 | 0.005 | 0.000 | 0.000 |
| | 1000B | 500B | 0.491 | 0.487 | 0.017 | 0.005 | 0.000 | 0.000 |
| 1.2B | 200B | 500B | 0.926 | 0.066 | 0.006 | 0.002 | 0.000 | 0.000 |
| | 300B | 500B | 0.758 | 0.229 | 0.012 | 0.001 | 0.000 | 0.000 |
| | 400B | 500B | 0.619 | 0.376 | 0.004 | 0.001 | 0.000 | 0.000 |
| | 500B | 500B | 0.619 | 0.376 | 0.004 | 0.001 | 0.000 | 0.000 |
| | 600B | 500B | 0.548 | 0.444 | 0.004 | 0.003 | 0.002 | 0.000 |
| | 700B | 500B | 0.496 | 0.492 | 0.007 | 0.003 | 0.002 | 0.000 |
| | 800B | 500B | 0.496 | 0.492 | 0.007 | 0.003 | 0.002 | 0.000 |
| | 900B | 500B | 0.496 | 0.492 | 0.007 | 0.003 | 0.002 | 0.000 |
| | 1000B | 500B | 0.496 | 0.492 | 0.007 | 0.003 | 0.002 | 0.000 |

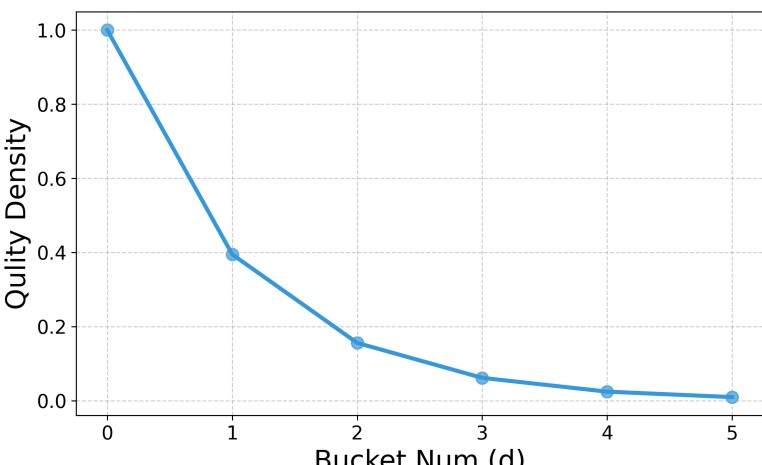

*Figure 10.* The fitted quality density function $f_d$ on the RefinedWeb dataset.

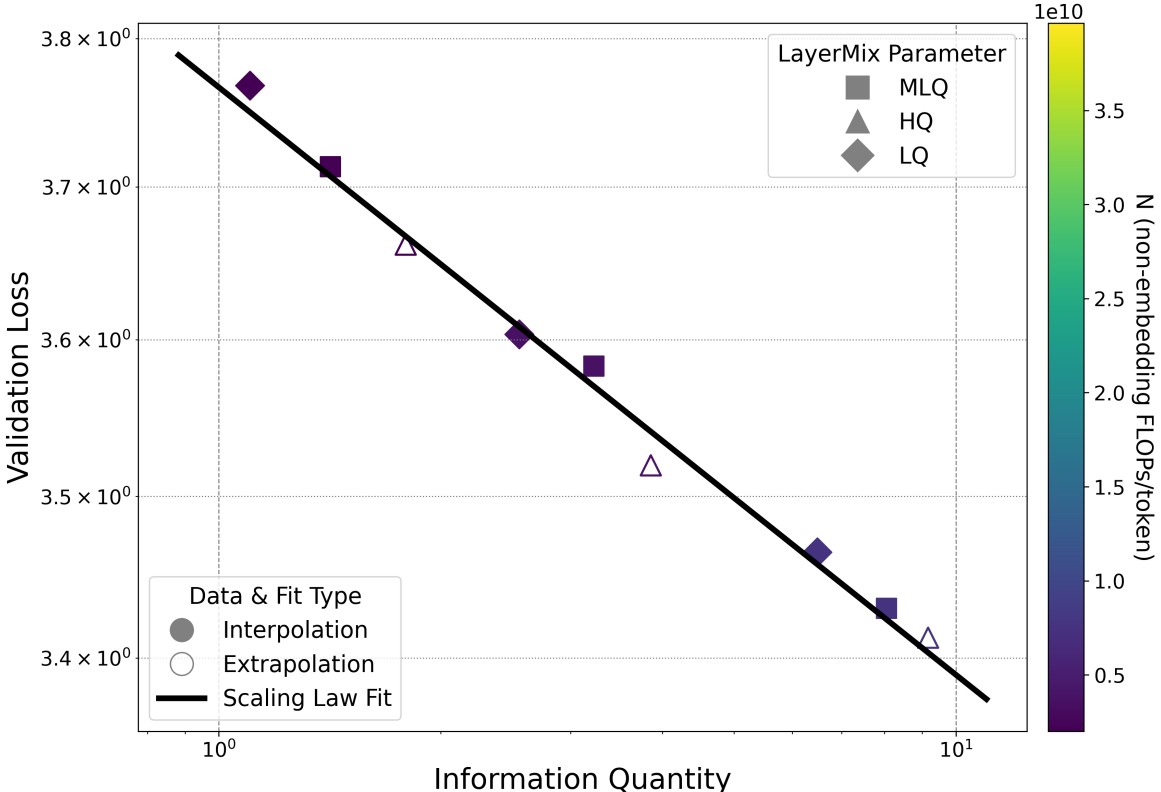

*Figure 11.* The Unified Information-Loss Scaling Law on the RefinedWeb dataset.

the curve along its intercept rather than rotating its slope.

*Table 9.* Fitted Info-loss power law parameters and accuracy across three overtraining factors. All curves share $f(\theta^*)$ and $\lambda(N)(a^*, b^*)$ fitted on $m = 3.6$. Errors are computed on out-of-fit experimental points within each regime.

| Regime | $\alpha$ | $-\beta$ | Mean abs. rel. err. | Max abs. rel. err. |
|---|---|---|---|---|
| $m = 1.7$ | 3.6909 | $-0.0465$ | 0.471% | 0.819% |
| $m = 3.6$ | 3.7373 | $-0.0441$ | 0.225% | 0.681% |
| $m = 25$ | 3.9664 | $-0.0463$ | 0.408% | 0.843% |

