# OpenReview forum: "InfoLaw: Information Scaling Laws for Large Language Models with Quality-Weighted Mixture Data and Repetition"
_ICML.cc/2026/Conference — ICML 2026 regular_

### Official Review · Reviewer_DEaj · 2026-02-22

**Soundness:** 2
**Presentation:** 1
**Significance:** 3
**Originality:** 3
**Overall Recommendation:** 4
**Confidence:** 4

**Summary:**

EDIT: After rebuttal I revised my paper score from 3 to 4.

The paper presents a new scaling law w.r.t. a new dimension: quality of data, its ratio, and its repetition. If some data is repeated in training, the scaling law will show a slower gain in performance (or even degradation). This is relevant in a data-limited regime as compared to the unlimited data case.

**Compliance With Llm Reviewing Policy:**

Affirmed.

**Final Justification:**

The paper has some merits stemming from the problem formulation. Some issues arise from lack of error bars and some weird notation that doesn't make sense in some parts. Most of my concerns are addressed in rebuttals.

**Key Questions For Authors:**

There are some complicated sounding phrases riddled throughout the paper, but are not defined anywhere or elaborated?
0. What is overtraining factor $m$? You mentioned it once in line 143 (right col) but did not define it. In addition, $m$ is used later in Table 1: "Optimal Recipe of 2.5B model with m= 3.6". We cannot judge what this table means without knowing what $m$ is. The bunch of variables from line 138-140 are also defined with magic numbers - *I suspect you are referencing these variables from another paper? Is that correct? If so I do not think that is a good practice at all.*
1. "We ran global fuzzy deduplication across all snapshots" - what does this mean?
2. Table 1 shows the "Searched optimal sampling weights" for different buckets of low quality to high quality data. What is this optimality w.r.t.? And how is it searched? Up until Table 1, I do not see any information about searched weights. EDIT: I found that it was referenced finally in the last page of the paper, when the Table is in page 4.
3. What does the sampling weight mean for the first 5 columns in Table 1?


I have other questions related to the problem formulation:
0. Eq (1) is strange: you mentioned that "Assuming the Information a document I contains is I_i, then the information a language model gets at t-th learning from the document I is: ..." - the resulting Eq (1) is supposed "to model the decreasing information gain of repeated data". But this function does not even take in any arguments that account for whether how much repeated data there is. Are we missing something here?
1. The next line says : "where λ(N) is a nonnegative rate parameter that depends on the model’s non-embedding FLOPs/token N and is fitted from data." - this phrase sounds informative but upon close inspection, I cannot understand this phrase at all.
2. "a language model gets at t-th learning from the document" - what is the t-th learning? t-th learning step? It is not elaborated as well. This makes it difficult to assess section 4.1 at all, since the whole section relies on t.

**Limitations:**

0. The paper's writing is very unclear and makes it difficult for a reader to understand. Many phrases are not defined and explained with a single sentence (see my questions to author). Many variables are *referenced from other papers without defining them*.
1. The problem formulation is unsound in some parts - from my understanding, they are trying to model how the performance of model scales with more data (but this data varies in mixing weights across different quality, and repeated different amount). But the information formulation (Eq 1) does not seem to incorporating the mixing weights or the repetition amount at all? The incorporating of these factors only appeared in Eq (5), and yet it seems to be incorporated in a fairly ad-hoc manner.
2. No standard deviation in the experimental plots - I think this is quite a big limitation. How far should we trust the results?
3. There some mathematical notation issues, like non-consistency of using subscripts and commas etc.

**Strengths And Weaknesses:**

0. The paper presents a new scaling law w.r.t. a new dimension: quality of data, its ratio, and its repetition. This idea is novel, and is original.
1. The idea is also significant because it considers a new dimension of scaling for scaling laws.

---

> ### Author Rebuttal · Authors · 2026-03-31
>
> **Notation and definitions**: "What is the overtraining factor? The variables around lines 138–140 appear to involve unexplained constants. What does 'global fuzzy deduplication' mean? Table 1's 'searched optimal sampling weights' and column meanings are unclear."
>
> We thank the reviewer for identifying these clarity issues.
>
> (1–2) Overtraining factor m and constants around lines 138–140: Both refer to the overtraining scaling relationships from Gadre et al. (2024). The overtraining factor quantifies how far training extends beyond the compute-optimal point: $K_m = \sqrt{m} \cdot K_{opt}$ and $N_m = \frac{1}{\sqrt{m}} \cdot N_{opt}$, so $m=1$ is compute-optimal and $m>1$ means the model is trained on more tokens with a smaller model than the Chinchilla-optimal pair for the same compute budget. We use $m=3.6$ throughout the fitting experiments.
>
> (3) Global fuzzy deduplication: This is MinHash-based near-duplicate detection (Lee, K., et al. 2022.Deduplicating Training Data Makes Language Models Better.) applied across all 96 Common Crawl snapshots, removing documents with high n-gram overlap.
>
> (4) Table 1 "searched optimal": This row shows the optimal data recipe identified by searching over 100k candidate LayerMix configurations using InfoLaw's predictions as described in Section 5.
>
> (5) Table 1 columns $w_0$–$w_5$: These are the sampling weights specifying the target token proportion from each quality bucket in the training data recipe. We will unify the terminology with Table 2 for consistency.
>
> We will revise these definitions and descriptions throughout the paper for standalone clarity.
>
> ---
> **Formulation clarity (Eq. 1, lambda(N), and "t-th learning")**: "Eq. (1) does not seem to take repetition as an explicit argument. The incorporation of those factors later in Eq. (5) appears ad hoc. What does 'the $t$-th learning from the document' mean? The description of $\lambda(N)$ is not sufficiently clear."
>
> We thank the reviewer for this question. To clarify:
>
> In Eq. (1), $t$ indicates the repetition times, it denotes the $t$-th time the document $i$ is exposed to the model , not a training step. The formulation models how the information gain from a single document decays exponentially with repeated exposure. When a document is seen $T$ times in total, Eq. (2) integrates over all $T$ exposures to give the total information learned from that document. In the full aggregation (Eq. 5), $T$ is replaced by the per-bucket average repetition factor $R_d = w_d K / M_d$, which is the average number of times each document in bucket $d$ is repeated during training, directly determined by the mixture weights $w$ and the source and training token budget, this is how repetition and mixture weights enter the formulation.
>
> $\lambda(N)$ is the decay rate controlling how quickly the gain diminishes. It increases with the model's non-embedding FLOPs/token $N$ because larger models extract information more efficiently, and therefore saturate faster on repeated data. The functional form $\lambda(N) = a \ln(N) + b$ is fitted from experimental data as detailed in Section 5.2, and we compare alternative functional forms in Appendix D, where the logarithmic form best fits the trend and extrapolates.
>
> The progression from Eq. (1) to Eq. (5) follows a natural bottom-up aggregation, not an ad hoc construction: Eq. (1) models a single exposure to one document; Eq. (2) integrates over all exposures; Eq. (5) sums across all quality buckets with their respective repetition factors and quality densities. We will restructure this derivation to make the progression more explicit.
>
> ---
> **Error bars and corrected metric**: "Experimental plots do not report standard deviations."
>
> Each configuration was evaluated using the last 3 checkpoints, and we report the mean across them to reduce checkpoint-level noise. Error bars were omitted from the figures for visual clarity. To further contextualize InfoLaw's prediction accuracy, we follow the Checkpoint-to-Checkpoint Noise framework from Belem et al. (2025), Signal and Noise: A Framework for Reducing Uncertainty in Language Model Evaluation, and computed the noise across all out-of-distribution evaluation points:
>
> | Metric | Mean | Max |
> |---|---:|---:|
> | InfoLaw absolute relative error | 0.262% | 0.96% |
> | Experimental relative SEM | 0.189% | 0.887% |
>
> All metrics above are computed on out-of-distribution points only, including unseen data recipes, unseen model scales, and their combinations. InfoLaw's mean prediction error is only 1.38× the experimental SEM, indicating that the predictions are approaching the measurement precision limit. We will add this analysis in the revision.
> We also note that the mean absolute error originally reported in the paper (0.15%) was incorrectly reported; the corrected value is 0.262%. We will update this in the revision.

---

> > ### Author Rebuttal · Reviewer_DEaj · 2026-04-03
> >
> > Thanks for the replies. My answers are mostly addressed. I will change my acceptance to weak accept (also ignore my deleted comments; I accidentally submitted this as a comment).

---

### Official Review · Reviewer_4LFP · 2026-02-26

**Soundness:** 2
**Presentation:** 2
**Significance:** 2
**Originality:** 2
**Overall Recommendation:** 4
**Confidence:** 3

**Summary:**

This work concentrates on building an information scaling law for LLMs with quality-weighted mixture data and repetition. The authors claim that high-quality data is rather limited in pre-training, and the trained tokens, model size, data mixture weights, and repetition are essential to the final pre-training loss. The built InfoLaw functions well on unseen data recipes.

**Compliance With Llm Reviewing Policy:**

Affirmed.

**Final Justification:**

I thank the authors for their rebuttal, which has addressed some of my questions. However, my concerns still exist: (a) considering the data processing and sense of real industrial level pre-training, the practicality of this infolaw needs to be discussed at the current stage. (b) Considering the rigor of scaling law, I suggest to use more experiment points and conduct further theoretically analysis to solidate this law. Currently, I will improve my rating from 2 to 3.

---

After Reply Rebuttal Comment:

For the pre-training data, I still believe that most of pre-training data should be carefully deduplicated and filtered. However, the third point of overtraining makes the proposed scaling law more practical. Maybe in the future, duplicated high-quality data will be more welcomed in pre-training.

For the number of experiment points, the number in this work is far less than those of other popular scaling laws (several hundred points). Moreover, evaluating with different overtraining ratio is also essential for infolaw.

I will increase the score to 4, and hope these discussions and limitations could be included in the final version.

**Key Questions For Authors:**

-	In Section 4, the equations are rather complex and have many hyper-parameters. How to determine the final formula from all possible selections? The author needs to provide a detailed introduction to the fitting formula.

-	The authors are suggested to validate the performance of the proposed InfoLaw compared to classical scaling laws on settings that have a relatively small over-trained ratio (which is more practical in industrial LLM pre-training).

**Limitations:**

Yes

**Strengths And Weaknesses:**

Strengths:

-	The proposed InfoLaw functions well on larger models with more trained tokens.

-	Extensive analyses have been conducted in the main content and appendix for a deeper understanding of this work.

Weaknesses:

-	The author needs to elaborate on the practicality of this scaling law under the current LLM pre-training paradigm. It seems that for most industry-level pre-training models, the used training data has very low repetition ratios. Moreover, Synthetic data is a mature choice for increasing (high-quality/diverse) training data.

-	The quality score is calculated via two quality classifiers. However, do the instances in six buckets share similar distributions on other essential attributes? Is there any natural bias present? For example, will code, math data, etc., appear more in a certain quality bucket? The authors should give more detailed analyses of this split.

-	In Section 5.1, the authors have conducted 27 experiment runs with 3.6x over-trained ratio to fit the InfoLaw parameters. Considering the complex form and hyper-parameters of InfoLaw, the number of experiments is too small for scaling law, making the conclusion less solid. Moreover, why do the authors merely select a fixed over-trained ratio?

---

> ### Author Rebuttal · Authors · 2026-03-31
>
> **Practicality and synthetic data**:
>
> We appreciate the observation, and in fact it aligns with our findings. As shown in Table 2, the optimal data recipe shifts toward lower quality emphasis but less repetition as model size increases. Because larger models have higher learning efficiency. For large industrial-scale models (>100B), the optimal strategy may indeed be to minimize repetition.
>
> Regarding synthetic data, it is a viable but costly approach, generating large-scale synthetic corpora requires strong teacher models, and ensuring sufficient diversity remains an open challenge. Our work focuses on a complementary and more cost-effective direction: optimally utilizing existing real data through principled quality-repetition trade-offs.
>
> The contribution of InfoLaw is a systematic study of how data quality and repetition affect model performance, and a framework that efficiently identifies the optimal data recipe for any given model size and compute budget, without exhaustive grid search over mixture configurations.
>
> ---
> **Bucket bias**:
>
> Our bucketing is performed within each domain independently, documents are ranked by quality score within their respective domain, then partitioned into buckets. LayerMix sampling from these buckets does not alter the original domain distribution, therefore does not introduce additional bias toward specific content types. We will clarify this procedure in the revision.
>
> ---
> **Fitting data sufficiency**:
>
> Despite the seemingly rich formulation, InfoLaw has only 5 free parameters in total: one for the quality density function ($\theta$), two for $\lambda(N)$ ($a, b$), and two for the info-loss power law ($\alpha, \beta$). The 27 training runs provide sufficient coverage for fitting. More importantly, the fitted model generalizes well: it predicts loss on unseen recipes and scales up to 7B model, and extrapolates reliably to a 25× overtraining regime.
>
> The overtraining ratio $m$ describes how many more tokens are used relative to the compute-optimal setting, and is not inherently tied to data repetition. Since our focus is on studying the effect of data quality and repetition, we fix $m=3.6$ to control for this confounding variable. The extrapolation to $m=25$ further validates that InfoLaw does not overfit to a specific overtraining ratio.
>
> ---
> **Formula selection**:
>
> Each component of the formula is motivated by specific assumptions and empirical observations:
>
> - Exponential decay form (Section 4.1): We assume information gain from repeated data decays exponentially, with the decay rate depending on model size.
>
> - Info-Loss power law (Section 4.2): We observe that information and loss form a straight line in log-log space, motivating the power-law relationship.
>
> - Quality density f and decay rate lambda (Section 5.2): We assume higher-quality buckets carry higher information density, and larger models have faster decay rates. For the quality density, we adopt a single-parameter exponential form$f_d = e^{-\theta d}$ to capture the monotonically decreasing information density. For the decay rate, we compared constant, power-law, and logarithmic forms for $\lambda(N)$ , with the logarithmic form performing best (Appendix B).
>
> We will provide a more detailed walkthrough of these design choices in the revision.
>
> ---
> **Comparison under low overtraining ratio**:
>
> We respectfully note that current industrial practice actually uses much higher overtraining ratios than ours. Qwen 3 trains a 32B model on 36T tokens, corresponding to roughly m=100, about 10× the compute-optimal token count (Gadre et al., 2024, Language Models Scale Reliably with Over-Training and on Downstream Tasks). Our fitting experiments use $m=3.6$, which is approximately 2× the compute-optimal amount.
>
> We also note that InfoLaw's core advantage is cross-recipe generalization: a single fitted model predicts loss across all data recipes by explicitly modeling quality and repetition, which traditional scaling laws do not account for.
>
> Nevertheless, we have conducted experiments at $m=1.7$ to validate InfoLaw under minimal overtraining:
>
> | Regime | $\alpha$ | $\beta$ | Avg Abs Error | Max Abs Error |
> |---|---:|---:|---:|---:|
> | $m=25$ | 8.24e12 | -0.04633 | 0.408% | 0.843% |
> | $m=1.7$ | 1.61e12 | -0.04646 | 0.471% | 0.820% |
>
> InfoLaw achieves comparable accuracy at $m=1.7$ (Avg Abs Error 0.471%) as at $m=25$ (0.408%). And the fitted exponent $\beta$ is nearly identical across all three regimes. The full figure will be included in the revision.

---

> > ### Author Rebuttal · Reviewer_4LFP · 2026-04-02
> >
> > I thank the authors for their rebuttal, which has addressed some of my questions. However, my concerns still exist: (a) considering the data processing and sense of real industrial level pre-training, the practicality of this infolaw needs to be discussed at the current stage. (b) Considering the rigor of scaling law, I suggest to use more experiment points and conduct further theoretically analysis to solidate this law.
> >
> > ---
> >
> > After Reply Rebuttal Comment:
> >
> > For the pre-training data, I still believe that most of pre-training data should be carefully deduplicated and filtered. However, the third point of overtraining makes the proposed scaling law more practical. Maybe in the future, duplicated high-quality data will be more welcomed in pre-training.
> >
> > For the number of experiment points, the number in this work is far less than those of other popular scaling laws (several hundred points). Moreover, evaluating with different overtraining ratio is also essential for infolaw.
> >
> > I will increase the score to 4, and hope these discussions and limitations could be included in the final version.

---

> > > ### Author Response · Authors · 2026-04-03
> > >
> > > (a) We respectfully disagree with the claim.
> > > 1. From our experience, after general cleaning and deduplication process, the common crawl data will remain about 10~20T.  Popular examples as Fineweb2 [1] 15T tokens. However, Qwen 3 [2] trained a 32B model on 36T tokens, Llama3 trained on 15T tokens. Those industrial practices generally train on larger than 15T tokens.
> > > 2. It is a general belief that data quality is more important than data quantity. There are so many works working on quality selection methods. After quality selection, the data will only remain less.
> > > 3. Current large language model generally believe that larger overtraining increases the inference performance (qwen3 36T is 10x tokens compared with the optimal scaling law), which will worsen the repetition.
> > > 4. Even under single epoch training strategy, the dataset is already produced by filtering and sampling to the target training tokens. The single epoch training doesn't necessarily mean there is no repetition.
> > >
> > > we are looking forward to the reviewer sharing some evidence or reference showing that the pretraining data only repeat once during the pretraining stage.
> > >
> > >
> > > (b) We have conducted sufficient experiment in order to prove the effectiveness the proposed InfoLaw.
> > > 1. training data points
> > >
> > > We conducted 9 settings with different computes (252M-1.2B model size) on 3 different data distributions, making 9x3=**27** experiments in total
> > >
> > > 2. validation data points
> > >
> > > We conducted
> > >
> > > a. 9 settings with different computes (252M-1.2B model size) on 2 hold-out data distributions (**18**)
> > >
> > > b. larger compute settings (1.4B-7B) on 5 preset data distributions (**12**)
> > >
> > > c. 25 unseen data distributions on 1.2B model (**25**)
> > >
> > > Our experiments have shown good extrapolations on all the validation data points. We have tested on three different kinds of extrapolations: unseen data distributions(a,c), larger computes (b), combination of larger computes and unseen data distributions(b)
> > >
> > > We have also further validated our InfoLaw on
> > >
> > > **1. different overtraining ratio**
> > >
> > > **2. different dataset**
> > >
> > >
> > >
> > > [1] FineWeb2: One Pipeline to Scale Them All — Adapting
> > > Pre-Training Data Processing to Every Language
> > >
> > > [2] Qwen3 Technical Report

---

### Official Review · Reviewer_Rgve · 2026-03-12

**Soundness:** 3
**Presentation:** 1
**Significance:** 3
**Originality:** 2
**Overall Recommendation:** 4
**Confidence:** 2

**Summary:**

This paper proposes InfoLaw (Information Scaling Laws) to predict model
performance by considering training tokens, model sizes, data mixing
ratios, and data repetition. It introduces an Information metric to
quantify the information gain a model obtains from a specific data
recipe. Using this metric, it fits a function to predict downstream
model performance. Experiments across 9 model sizes and diverse dataset
configurations verify the effectiveness of this framework, showing
extrapolation ability across overtraining levels, unseen data recipes,
and unseen model sizes.

**Compliance With Llm Reviewing Policy:**

Affirmed.

**Final Justification:**

I maintain my score of 4. The authors have addressed my concerns regarding presentation and experimental setup. I want to be transparent that data mixing for large-scale pretraining is not my area of expertise, so I encourage the area chair to weigh the assessments of more domain-expert reviewers accordingly.

**Key Questions For Authors:**

See Weaknesses.

**Limitations:**

Limitations of this work should be discussed.

**Strengths And Weaknesses:**

Strengths:
- How to select the optimal data recipe for model training at scale is of practical interest, as it allows us to improve training efficiency with minimal extra cost.

- The motivation is intuitive and clear: after repetition, the benefit of high-quality data diminishes and approaches that of unseen low-quality data, which necessitates modeling both data repetition and data quality.

- Comprehensive experiments are conducted to support the motivations behind the framework and verify its effectiveness. The results show that the proposed framework achieves better prediction results than conventional scaling laws and demonstrates the ability to extrapolate across overtraining levels, unseen data recipes, and unseen model sizes.

- Experiments on a 2B model show that the optimal data recipe searched using this framework can achieve better model performance than other fixed recipes, showing its practical utility.


Weaknesses:
- The writing of this paper needs improvement. It introduces too many symbols, requiring readers to constantly track their meanings. A symbol table would help make the paper easier to follow. Additionally, some figures also need to be reformatted: Figure 4 spans two columns and should be condensed to one column, and the subfigures in Figure 3 are too small to read.

- For Figure 1, it is unclear why the authors do not consider other settings such as MQ and MLQ.

- The preset LayerMix sampling weights for datasets of different quality look quite arbitrary. What is the motivation behind the selection of these sampling weights? Are the specific weight values related to any practical scenario?

- In current practice, models are often pretrained for a single epoch, meaning there is no data repetition in the training recipe. It is unclear whether the framework remains effective in this setting. Additionally, what is the cost of assessing data quality?

---

> ### Author Rebuttal · Authors · 2026-03-31
>
> **Presentation (notation, figures, symbol table)**: "The writing introduces too many symbols... A symbol table would help. Figure 4 should be condensed to one column. The subfigures in Figure 3 are too small to read. Figure 1 does not explain why settings such as MQ and MLQ are omitted."
>
> Thank you for the detailed presentation feedback. In the revision, we will: (1) add a symbol table summarizing all key notation; (2) resize Figure 3 subfigures, increase font sizes for readability, and condense Figure 4 to a single column; (3) add a note in Figure 1 explaining that MQ and MLQ are omitted for visual clarity, with the full set of curves provided in Appendix Figures 8 and 9.
>
> ---
> **Preset sampling weights**: "The preset LayerMix sampling weights across quality levels appear arbitrary."
>
> The five preset LayerMix weights were not chosen arbitrarily. We first randomly generated 30 candidate weight configurations, then selected 5 that span a range of repetition levels — HQ(4.35×), MHQ (3.33×), MQ (2.13×), MLQ (1.69×), and LQ (1.35×). This design ensures the fitting process is exposed to diverse mixing strategies covering a range of repetition levels. The effectiveness of this selection is validated empirically: InfoLaw fitted on only three of these presets (HQ, MQ, LQ) generalizes well to the remaining held-out presets (MHQ, MLQ) and to 25 additional randomly sampled weight configurations on the 1.2B model. We will clarify this selection procedure in the revision.
>
> ---
> **No-repetition applicability**: "In current practice, many models are pretrained for only one epoch, so repetition may not occur. It is unclear whether the framework remains effective in the no-repetition setting."
>
> We consider two interpretations of this question:
>
> (1) Single epoch with truly no repetition — all data is uniformly sampled without upweighting. In this case there is no quality-repetition trade-off, and the problem reduces to standard scaling, where traditional scaling laws already work well.
>
> (2) Single epoch with quality-weighted sampling — high-quality subsets are upweighted beyond their natural proportion, which inherently induces repetition in those subsets even within one epoch. This is exactly the scenario InfoLaw addresses, and is common in practice when practitioners curate training mixtures to prioritize higher-quality data.
>
> We believe scenario (2) is more representative of current practice, as most pipelines apply some form of quality-based filtering or upweighting.
>
> ---
> **Quality assessment cost**: "The cost of assessing data quality is not discussed."
>
> The quality assessment cost is low. We use a fastText classifier as the DCLM scorer, and a distilled BERT model as the FineWebEdu scorer. Scoring the entire corpus takes approximately 16 hours on 64 H100 GPUs — negligible compared to the pretraining compute. We will add this detail in the revision.
>
> ---
> **Limitations**: "Limitations should be discussed explicitly."
>
> We thank the reviewer for raising this point, which was also noted by Reviewer WiCz. We will move the key limitations from Appendix I into the main-text conclusion. Please see our response to Reviewer WiCz for a more detailed discussion.

---

> > ### Author Rebuttal · Reviewer_Rgve · 2026-04-01
> >
> > Thank you for the clarifications. My concerns are resolved, and I will maintain my positive score.

---

### Official Review · Reviewer_WiCz · 2026-03-13

**Soundness:** 3
**Presentation:** 4
**Significance:** 4
**Originality:** 3
**Overall Recommendation:** 5
**Confidence:** 3

**Summary:**

This paper studies how to determine optimal data mixtures for LLM training through scaling laws. Transferring optimal data mixtures across scales is challenging since high quality data is limited, and its repetition breaks scaling law trends. The authors design a scaling law, based on treating pretraining as information accumulation, that accounts for the repetition and quality of the training data when modeling loss across scales.

**Compliance With Llm Reviewing Policy:**

Affirmed.

**Final Justification:**

The paper proposes InfoLaw, a scaling law that accounts for data repetition and quality when modeling LLM training loss across scales. Planning effective data mixtures is a relevant problem for large-scale language model pretraining.

The authors addressed my main concerns satisfactorily. In particular, they clarified that the conventional scaling law was re-fitted on repeated data (rather than transferred from a repetition-free setting), strengthening the claim that standard power laws are insufficient under repetition. The cross-corpus discussion was also helpful and highlights an important direction for future work.

Overall, the rebuttal reinforced my positive assessment, and I maintain my score of 5.

**Key Questions For Authors:**

**Q1:** In the section “Limitations of Conventional Scaling Laws”, the experiments follow the Chinchilla prescription for scaling model size and training tokens. What is the effect of fixing this scaling trajectory when data repetition occurs? If the Chinchilla fitting procedure were repeated under these conditions, it might yield a different optimal relationship between model size and training tokens. Do the authors believe that refitting this relationship could account for the poor extrapolation performance observed for the conventional scaling law?

**Q2:** The quality density function $f_d(\theta) = e^{-\theta d}$ depends on the bucket index $d$, which reflects the ordering of data by quality within the source corpus. How easily do the authors expect this functional form to transfer to other settings where the datasets may come from different corpora?

**Limitations:**

The authors acknowledge that the choice of quality buckets is heuristic and that the effect of the overtrain degree lacks a theoretical explanation. The discussion of limitations could be strengthened by addressing how the proposed procedure might transfer to datasets drawn from different corpora.

**Strengths And Weaknesses:**

**Soundness**
The authors clearly motivate the design of InfoLaw and empirically demonstrate its ability to accurately model downstream performance when extrapolating across model scales and training token counts or changing dataset mixtures.

In Section 3, the authors argue that Figure 1 shows that “a conventional power-law scaling curve […] systematically mis-extrapolates” when data is repeated. However, this claim is not conclusively demonstrated. The experiments follow the Chinchilla scaling prescription for model size and training tokens when generating the data used to fit the scaling law.
If the Chinchilla fitting procedure were repeated in this setting, it might yield a different optimal scaling of model size and training tokens, allowing for more accurate extrapolation.
Given the ambiguity regarding the cause of the observed deviation, conclusions about the failure of conventional scaling laws should be stated more cautiously.

Additionally, there appear to be some inaccuracies in the attribution of related work.

The paper states:

> Empirical studies have shown that transformer language models exhibit predictable power-law scaling with model size and training data (Hestness et al., 2017; Vaswani et al., 2017; Chowdhery et al., 2023; Radford et al., 2019)

However, Vaswani et al. (2017) introduce the Transformer architecture and do not study scaling-law relationships.

The paper also states:

> Sardana et al. (2024) extended the Chinchilla framework by incorporating factors such as data quality and inference requirements,

To my understanding, Sardana et al. (2024) consider inference costs when deriving optimal training regimes but do not model data quality. Clarifying these citations would improve the accuracy of the related work discussion.

**Presentation**

The narrative is well-crafted and easy to follow. However, the Conclusion section does not discuss the work's limitations. Since there is a limitation section in the appendix, it would improve the presentation if the conclusion referred to it.

**Significance**

This paper studies how to determine optimal data mixtures for LLM pretraining by proposing a scaling law that provides guidance on how data of different quality levels should be mixed. Data mixture design is an important problem in modern LLM training, making this work relevant to the venue.

**Originality**

This work is fairly original. It proposes a new functional form for a scaling law that models the effect of data mixtures during pretraining. The formulation also provides guidance on how repeated exposure to data should be handled, which, to the best of my knowledge, is a novel contribution.

---

> ### Author Rebuttal · Authors · 2026-03-31
>
> **Conventional scaling law under repetition**: "If the Chinchilla fitting procedure were repeated under repetition conditions, might it account for the poor extrapolation performance of the conventional scaling law?"
>
> Thank you for this important question. We would like to clarify that the traditional scaling law reported in our paper was indeed re-fitted on data with repetition, not transferred from a repetition-free setting. Specifically, in Figure 1, the conventional power-law curve $L = a \cdot C_m^{-b}$ is fitted on models from 252M to 1.2B trained on the same HQ and MLQ datasets that contain repetition, where the top-5% bucket in HQ is repeated roughly 16×. And the traditional scaling law still systematically under-estimates loss when extrapolating to larger compute such as 2.5B and 7B, yielding overly optimistic predictions.
>
> Appendix Figure 9 further confirms this: across all five LayerMix configurations where repetition is induced, the per-recipe power-law fitted on the smallest three model scales consistently deviates from the actual loss at larger scales.
> These results suggest that the observed extrapolation failure is not simply caused by mismatched fitting conditions. Instead, in our repeated-data setting, the standard L-C power law appears insufficient to capture the diminishing returns of repeated data, especially when extrapolating to larger model scales. InfoLaw is designed to address this issue by replacing the compute axis with a repetition- and quality-aware information metric.
>
> We will revise Section 3.2 to state more explicitly that the traditional law is re-fitted on each recipe's own repeated data.
>
> ---
> **Cross-corpus transferability**: "How well do the authors expect the quality density function to transfer to settings where datasets come from different corpora?"
>
> We have already conducted a cross-corpus verification on RefinedWeb (Appendix H). Despite different filtering pipelines, the fitted $\theta$ on RefinedWeb is 0.93, remarkably close to our primary dataset's 0.92, and InfoLaw achieves 0.24% mean / 0.36% max absolute error on held-out recipes. This suggests the quality density function captures a genuine quality–information relationship rather than overfitting to a specific dataset.
>
> ---
> **Related-work concerns**: "Vaswani et al. (2017) introduced the Transformer architecture and did not study scaling-law relationships. Sardana et al. (2024) considered inference costs but did not model data quality."
>
> Thank you for pointing out these inaccuracies. We have revised the Related Work section accordingly. Specifically, we corrected the citation of Vaswani et al. (2017) to properly attribute it as the Transformer architecture, and revised the description of Sardana et al. (2024) to: "Sardana et al. (2024) extended the Chinchilla framework by incorporating inference costs when deriving optimal training regimes."
>
> ---
> **Limitations in conclusion**: "The conclusion does not discuss limitations, even though there is a limitations section in the appendix."
>
> We agree. In the revision, we will add a limitations discussion to the main-text conclusion. Regarding the effect of overtraining degree: when the data mixture and repetition are held constant, overtraining still follows the traditional scaling law, as shown by the Random curve in Appendix Figure 7(a). However, under repetition, the traditional scaling law breaks down (Appendix Figure 9). A theoretical explanation for why the overtraining degree systematically shifts the InfoLaw curve remains an open question, which we will acknowledge as a limitation.

---

> > ### Author Rebuttal · Reviewer_WiCz · 2026-04-03
> >
> > Thank you for clarifying that the Chinchilla laws were refit and not simply transferred.
> >
> > I want to clarify my question about cross-corpus transferability. In the paper's setting, a single corpus is split into ordered buckets based on document quality.
> > For example, in SmolLM2 (Allal et al., 2025), the mixed datasets represent different domains: English web, math, coding, and textbooks.
> > In this setting, it might not be straightforward to use a quality density function, since even if the datasets can be ordered by quality, the indices would be ordinal rather than quantitative, limiting the application of InfoLaw.
> > Can InfoLaw be adapted to this setting?
> >
> > **References**
> > Allal, Loubna Ben, et al. "SmolLM2: When Smol Goes Big--Data-Centric Training of a Small Language Model." arXiv preprint arXiv:2502.02737 (2025).

---

> > > ### Author Response · Authors · 2026-04-03
> > >
> > > Thanks for the further clarifying. In this paper we already consider the effect of domain mixture in a single corpus. Specifically, we do not rank the whole corpus based on the quality score. Instead, we first split the dataset based on the domains, and then rank the data within one domain, then we split the bucket within one domain. Finally, we group the data from all domains in the same bucket index together to get the bucketed data. We will further clarify this in the future version.
> > >
> > > What this can achieve:
> > > 1. the indices is indeed **quantitative**, not just ordinal. If an example is in bucket1, that means it is ranked top5% within the domain it belongs to.
> > > 2. the data mixture is not changing no matter how we change the sampling strategy (changing the layermix sampling parameters), reducing the effect of mixture changing to the model performance.
> > > 3. the quality density function is reflecting the average affect from a fixed domain mixture.
> > >
> > > We further tested how the our theory can transfer to other corpus where the mixture is different.(Appendix K) We run similar experiment on RefineWeb and fit the parameters under the same Infolaw formulation and we can still get good fitting results, meaning that the assumption of Infolaw is invariant to the data mixture.
> > >
> > > It would be more interesting to explore a unified scaling law theory to also include the influence of domain mixture under quality selection and repetition. We will add discussion for this in the future work.

---

### Decision · Program_Chairs · 2026-04-30

**Decision:**

Accept (regular)

**Comment:**

This paper studies more fine-grained scaling laws for training large language models. The idea is to integrate data aspects into the scaling laws (rather than the traditional version which just models data quantity). In particular, the authors produce scaling laws that are quality (and repetition) aware, which is new and useful.

Reviewers had a positive impression of the paper (and I largely agree with them). The basic tradeoff being studied here (wanting to repeat high-quality data but only to an extent) is interesting, and the mechanisms proposed are reasonable. The work is going to be practically useful as well.

The main concerns are mostly about writing and clarity; the authors did a good job in responding about these but the final version could definitely use some cleanup. The other concern that comes up is practicality, but the authors have clarified and it is likely that the work really is going to be useful. Additionally I do think the authors should do a better job contextualizing their work—there are uncited efforts that are also data-aware (see Shukor et al “Scaling Laws for Optimal Data Mixtures”—more focused on data domains). The authors do differentiate on the basis of being repetition-aware, which is reasonable, but still getting a complete picture would be great.

Overall this is a solid work with valuable contributions.